# PROVABLE STRATEGIC IN-CONTEXT LEARNING OF TRANSFORMERS

## ABSTRACT

In-context reinforcement learning (ICRL) enables Transformers to adapt to new decision-making tasks using context alone without parameter updates. Recent works demonstrate that Transformers can replicate reinforcement learning strategies, such as V-learning, in uncoupled learning environments. This provides valuable insights and lays the theoretical foundation for applying Transformers to strategic settings. However, their approach relies on external sampling mechanisms during inference, introducing an artificial layer on top of the original Transformer structure. This work investigates whether Transformers can perform in-context game playing in matrix and Markov zero-sum games entirely within their architecture, without external procedures. We show the first theoretical result that Transformers can approximate online mirror descent (OMD) dynamics in both repeated and sequential multi-agent games, with the accuracy dependent on model size. Additionally, we demonstrate the benefit of pre-training with longer trajectories under appropriate model architecture choices.

## 1 INTRODUCTION

In-context reinforcement learning (ICRL) refers to a Transformer model's ability to learn and adapt to new sequential decision-making tasks purely through context, without any parameter updates Laskin et al. (2023); Lee et al. (2023); Lu et al. (2023). After being pre-trained on diverse interaction trajectories, the Transformer is prompted at inference time with a short history from a new environment and is expected to select actions effectively based on this context alone. This paradigm enables a single, frozen model to generalize across tasks. The appealing properties of this paradigm have therefore motivated many theoretical and empirical studies to better understand in-context learning Xie et al. (2022); Garg et al. (2022); Akyürek et al. (2023); Von Oswald et al. (2023); Dai et al. (2023); Bai et al. (2024).

One conjecture of how Transformer conduct in-context learning is it does so by implicitly implementing learning algorithms within its forward pass. In fact, recent work has formalized in-context reinforcement learning (ICRL), demonstrating that well-designed Transformers can replicate classic algorithms such as LinUCB and Thompson Sampling in single-agent settings like bandits and Markov decision processes Lin et al. (2024). These findings suggest that Transformers, when pre-trained on appropriate distributions of tasks, can internalize exploration and learning dynamics purely through context.

However, this line of research has so far focused largely on isolated, single-agent environments. In contrast, recent advances such as AlphaProof AlphaProof & AlphaGeometry (2024) and DeepSeek Liu et al. (2024) highlight the growing importance of interactive experience, where models continually improve by generating and learning from their interactions. Many of these interactive settings are inherently multi-agent, involving coordination, competition, negotiation, and collaboration among intelligent agents. In such environments, progress relies not only on adapting to a dynamic world but also on reasoning about and responding to the behavior of other agents. This elevates agent-to-agent strategic interaction as a central challenge. Extending ICRL to these game-theoretic settings, known as in-context game playing, introduces new theoretical and algorithmic challenges.

Shi et al. Shi et al. (2024) expands beyond single-agent settings to investigate Transformers' game-playing abilities under the uncoupled learning paradigm. Specifically, two independent Transformers are pre-trained separately, each without access to the other's actions. The work show that during

inference time, the two Transformers can find an approximate Nash equilibrium (NE) by realizing the V-learning algorithm Jin et al. (2024). This proves that Transformers can encode reinforcement learning strategies directly in context.

However, a key limitation of the previous work lies in its reliance on an external sampling mechanism during inference. In fact, realizing V-learning in the uncoupled setting requires a sophisticated averaging procedure to simulate stochastic mixtures of historical policies. Effectively, it performs a weighted sampling over past iterates. This introduces an artificial inference-time component outside the Transformer's architecture. This raises a natural question:

**Can a Transformer, without any external mechanisms, learn to play games purely in context?**

We start the investigation of this question with competitive games. In this work, we study the in-context game-playing capabilities of Transformers in both matrix games and Markov games under an uncoupled learning setting, focusing on whether the model can implement no-regret learning dynamics internally. Our main results are as follows.

1. We provide the first theoretical results showing that Transformers can perform in-context learning in competitive games without relying on any external sampling algorithm at inference time. We further validate this with empirical results.

2. Technically, we develop a new approach to analyze Transformers. We show that Transformers can learn in-context by approximating online mirror descent (OMD) dynamics in repeated normal-form games and extend this capability to sequential multi-agent interactions in Markov games. To enable this, we reformulate the constrained OMD updates using the Lagrangian method, allowing Transformers to approximate the resulting unconstrained optimization dynamics. Instead of relying on smooth or approximation through gradient descent, our approach bridges constrained dynamics and transformer architectures through a careful analysis of the smooth surrogate objective and its approximation error.

3. Our results show that while Transformers can approximate OMD up to an arbitrary $\epsilon$-accuracy, doing so imposes architectural constraints. Specifically, the achievable accuracy depends on the model size, revealing a fundamental tradeoff between the model capacity and the strategic learning ability.

4. We extend the results to Markov zero-sum games. While the previous error bound scales linearly with the trajectory length $T$, which suggests a less efficient training with longer trajectories, our results show the opposite that the duality gap in in-context game playing decreases as $T$ increases, provided the Transformer architecture is sufficiently large.

## 2 RELATED WORKS

**Provable In-context Learning**   In-context learning (ICL) with transformer models has garnered significant attention in recent years, fueled by advancements in large language models (LLMs). A growing body of research aims to understand the fundamental mechanisms behind ICL. Several studies have demonstrated that transformers can perform in-context gradient descent, enabling the implementation of various optimization-based algorithms Garg et al. (2022); Akyürek et al. (2023); Von Oswald et al. (2023); Ahn et al. (2023). Looped transformers have also been shown to emulate essential computational primitives, which can be combined to execute more complex operations Giannou et al. (2023).

Recent work by Lin et al. Lin et al. (2024) established generalization guarantees for pre-trained models in the context of single-agent reinforcement learning. They proposed transformer architectures that provably implement efficient algorithms for single-agent bandits and reinforcement learning. A follow-up study Shi et al. (2024) extended this research to multi-agent settings, illustrating the in-context game-playing capabilities of transformers. This was demonstrated by the transformers' ability to realize the efficient multi-agent learning algorithm, V-learning Jin et al. (2024). However, since V-learning requires stochastic sampling of history iterates, the in-context game-playing ability of transformers is limited without an external sampling algorithm.

**Learning in (Markov) Zero-sum Games**   For constrained zero-sum games with a unique Nash equilibrium, Daskalakis & Panageas (2019) established asymptotic last-iterate convergence for the

optimistic multiplicative weight update method. This result was later improved by Wei et al. (2020), who demonstrated a linear last-iterate convergence rate. However, their analysis depends on problem-specific parameters, such as a condition-number-like quantity. Subsequent works explored optimistic learning algorithms with vanishing learning rates Mertikopoulos et al. (2019), as well as extensions to more complex variants of zero-sum games, including Markovian settings Daskalakis et al. (2020); Wei et al. (2021); Yang & Ma (2022); Leonardos et al. (2022). Recent work Cai et al. (2024) also showed last-iterate convergence properties of online mirror descent-based algorithms in (Markov) zero-sum games under bandit feedback.

## 3 PRELIMINARY

### 3.1 TWO-PLAYER (MARKOV) ZERO-SUM GAME

**Two-player Zero-sum Game**  Consider a set of two-player zero-sum games $\mathcal{G}$, where each game $G \in \mathcal{G}$ shares action spaces $\mathcal{A}, \mathcal{B}$. Each game $G$ has a reward function $R_G : \mathcal{A} \times \mathcal{B} \to \mathbb{R}$. With a joint action $a \in \mathcal{A}, b \in \mathcal{B}$, the $a$-player receive a reward of $r_G(a, b) \sim R_G(a, b)$, and the $b$-player receive a reward of $-r_G(a, b)$. We thus call the $a$-player the max player and the $b$-player the min player. Without loss of generality, we assume $r_G(\cdot, \cdot) \in [0, 1]$, for any $G \in \mathcal{G}$ and follow the form of $a^\top A_G b$, where $A_G$ is a game specific matrix. The players can also choose to play randomized policies $x \in \Delta(A), y \in \Delta(B)$, and $r_G(x, y) = \mathbb{E}_{a \sim x, b \sim y}[r_G(a, b)]$. A pair of policies $(x^*, y^*)$ is said to be a Nash equilibrium in a game $G$ if for any $x, y$, $(x^*)^\top A_G y \le (x^*)^\top A_G y^* \le x^\top A_G y^*$. The minimax theorem guarantees the existence of a Nash equilibrium in a two-player zero-sum game (v. Neumann, 1928).

**Two-player Markov Zero-sum Game**  Consider a set of two-player Markov zero-sum games $\mathcal{G}$, where each of the games $G \in \mathcal{G}$ shares a state space $\mathcal{S}$, and action spaces $\mathcal{A}, \mathcal{B}$. Each $G$ has a transition function $T_G : \mathcal{S} \times \mathcal{A} \times \mathcal{B} \to \mathcal{S}$, a reward function $R_G : \mathcal{S} \times \mathcal{A} \times \mathcal{B} \to \mathbb{R}$ and a discount factor $\gamma$. In a zero-sum game $G$, upon state $s \in \mathcal{S}$ and joint action $a \in \mathcal{A}, b \in \mathcal{B}$ the $a$-player will receive a reward of $r_G(s, a, b) \sim R_G(s, a, b)$ whereas the $b$-player will receive a reward of $-r_G(s, a, b)$. Without loss of generality, we assume $r_G(\cdot, \cdot, \cdot) \in [0, 1]$, for any $G \in \mathcal{G}$. The players can also choose to play randomized policies $x \in \Delta(A), y \in \Delta(B)$.

The expected payoff (value function) for the max player in a game $G$ is $V_{x,y}^{G,s} = \mathbb{E}\left[\sum_{t=1}^{\infty} \gamma^{t-1} r_G(s_t, a_t, b_t) \mid s_t \sim T_G(s_{t-1}, a_t, b_t), a_t \sim x, b_t \sim y, s_0 = s\right]$, and the expected payoff of the min player is the negative of the reward. The Q-value of a game $G$ under state-action $s, a, b$ and policy $x, y$ is $Q_{x,y}^{G,s}(a, b) = r_G(s, a, b) + \mathbb{E}\left[V_{x,y}^{G,s'} \mid s' \sim T_G(s, a, b)\right]$. When we rewrite this as a matrix $Q_{x,y}^{G,s}$, we can see that $V_{x,y}^{G,s} = x^\top Q_{x,y}^{G,s} y$. We denote the minimax game value on state $s$ as $V_*^s = \max_x \min_y V_{x,y}^s$. We call a policy $z^* = (x^*, y^*)$ a Nash equilibrium if it attains the minimax game value.

**Bandit Feedback and Uncoupled Learning Dynamics**  In this paper, we consider games under bandit feedback and uncoupled learning dynamics. In this setting, each player chooses their policies individually and does not observe others' actions. Both players can only observe the reward that depends on both the actions chosen.

### 3.2 IN-CONTEXT LEARNING

In this section, we introduce the setting for in-context learning for two-player zero-sum games.

**Distribution of Offline Dataset**  We define a partial interaction trajectory for the max player at time $t$ as $D_{x,t} = \{(a_1, r_1), \ldots, (a_t, r_t)\}$, and analogously for the min player as $D_{y,t} = \{(b_1, r_1), \ldots, (b_t, r_t)\}$. In a Markov zero-sum game setting, these trajectories also include state information. Specifically, for the max player, the trajectory becomes $D_{x,t} = \{(s_1, a_1, r_1), \ldots, (s_t, a_t, r_t)\}$ and for the min player, $D_{y,t} = \{(s_1, b_1, r_1), \ldots, (s_t, b_t, r_t)\}$.

We focus primarily on notation for the max player; analogous definitions apply for the min player. A max-player algorithm $\mathsf{Alg}_x$ maps a partial interaction trajectory $D_{x,t-1}$ to a distribution over actions $\Delta(A)$. Given a game instance $G$ and an algorithm $\mathsf{Alg}_x$, the distribution over

a trajectory $D_{x,T}$ is defined as follows: 1) In a (non-Markov) zero-sum game: $P_G^{\mathsf{Alg}_x}(D_T) = \prod_{t=1}^T \mathsf{Alg}_x(a_t \mid D_{t-1})R_G(a_t, b_t)$, 2) In a Markov zero-sum game: $P_G^{\mathsf{Alg}_x}(D_T) = \prod_{t=1}^T \mathsf{Alg}_x(a_t \mid D_{t-1})R_G(s_t, a_t, b_t)$.

Let $\Lambda \in \Delta(\mathcal{G})$ denote a prior distribution over games. Then, we write $P_G^{\mathsf{Alg}_x, \Lambda}$ to denote the joint distribution over a randomly drawn game $G \sim \Lambda$ and the resulting trajectory $D_T$ induced by $\mathsf{Alg}_x$.

**Supervised Pre-training** During supervised pre-training, we employ a context algorithm $\mathsf{Alg}_c = (\mathsf{Alg}_{x,c}, \mathsf{Alg}_{y,c})$ to collect offline datasets $D_{x,T}$ and $D_{y,T}$ for the max and min players, respectively. For each trajectory $D_{x,T}$, we augment it with actions recommended by an expert algorithm $\mathsf{Alg}_{E,x}$. The augmented trajectory is denoted by $\bar{D}_{x,T} = D_{x,T} \cup \{\bar{a}_t\}_{t=1}^T$, where $\bar{a}_t \sim \mathsf{Alg}_{E,x}(\cdot \mid D_T)$ in the zero-sum case and $\bar{a}_t \sim \mathsf{Alg}_{E,x}(\cdot \mid D_T, s_T)$ in the Markov zero-sum case. The resulting distribution over the augmented trajectory $\bar{D}_{x,T}$ is given by $P_G^{\mathsf{Alg}_x, \mathsf{Alg}_{E,x}}(D_T) = P_G^{\mathsf{Alg}_x}(D_T) \prod_{t=1}^T \mathsf{Alg}_{E,x}(a_t \mid D_{x,T})$, or $P_G^{\mathsf{Alg}_x, \mathsf{Alg}_{E,x}}(D_T) = P_G^{\mathsf{Alg}_x}(D_T) \prod_{t=1}^T \mathsf{Alg}_{E,x}(a_t \mid D_{x,T}, s_T)$ in the case of Markov game.

For pre-training, we first sample $n$ offline trajectories $\{D_{x,T}^i\}_{i=1}^n$ from the joint distribution $P_G^{\mathsf{Alg}_x, \Lambda}$. We then augment each with expert actions $\{\{\bar{a}_t^i\}_{t=1}^T\}_{i=1}^n$. The training objective is to maximize the log-likelihood over a parameterized algorithm class $\mathsf{Alg}_{\theta,x}$ by solving the following optimization problem:

$$\hat{\theta} = \arg\max_\theta \frac{1}{n} \sum_{i=1}^n \sum_{t=1}^T \log\left(\mathsf{Alg}_{\theta,x}\left(\bar{a}_t^i \mid D_{x,t-1}^i\right)\right), \tag{1}$$

or $\hat{\theta} = \arg\max_\theta \frac{1}{n} \sum_{i=1}^n \sum_{t=1}^T \log\left(\mathsf{Alg}_{\theta,x}\left(\bar{a}_t^i \mid D_{x,t-1}^i, s_t^i\right)\right)$ in the case of Markov zero-sum game.

**Learning Objective** In two-player zero-sum game, a pair of policies $(x, y)$ is said to be an $\epsilon$-approximate Nash equilibrium in a game $G$ if for any $x', y'$, $(x)^\top A_G y' \leq (x)^\top A_G y + \epsilon$, $(x')^\top A_G y \geq (x)^\top A_G y - \epsilon$. Equivalently, this means that neither player can gain more than $\epsilon$ by unilaterally deviating from their strategy. Similarly, in a Markov two-player zero-sum game, a policy pair $z = (x, y)$ is said to be an $\epsilon$-approximate Nash equilibrium in a Markov zero-sum game if for all states $s$ and all alternative policies $x'$ and $y'$, $V_{x,y'}^s \leq V_{x,y}^s + \epsilon$, $V_{x',y}^s \geq V_{x,y}^s - \epsilon$, $\forall s$. For each environment, our learning goal is to approximate its NE policy pair. In other words, we target outputting a policy pair that is $\epsilon$-approximate NE with an error $\epsilon$ that decreases as $T$ increases, after interacting with the environment for an overall $T$ rounds.

### 3.3 Algorithms Induced by Transformers

With the background and objective of in-context learning established, we now introduce the transformer architecture that will implement our algorithm Vaswani et al. (2017).

**Embedding Mappings** The max and min players have two embedding mappings $h_x : \mathcal{A} \times \mathcal{O} \to \mathbb{R}^{d_x}$ and $h_y : \mathcal{B} \times \mathcal{O} \to \mathbb{R}^{d_y}$ with dimensions $d_x$ and $d_y$ respectively. Here, $\mathcal{A}$ represents the action space for the max player, $\mathcal{B}$ represents the action space for the min player, and $\mathcal{O}$ represents the observation space (which include rewards and state information if the game is Markov, and only rewards if the game is non-Markov).

Next, we define the algorithm induced for the max player, with similar definitions applying to the min player. The embedding mapping sends any action-observation pair into an embedding, creating a sequence of embeddings for the historical data:

$$h(D_{x,t-1}) = [h_x(a_1, o_1), \dots, h_x(a_{t-1}, o_{t-1})], \tag{2}$$

where $D_{x,t-1}$ represents the history of the max player's actions and the corresponding observations up to time $t-1$.

**Transformer Processing** The transformer $\mathrm{TF}_{x,\theta}$ processes these embeddings to produce an output with the same shape. To extract the action distribution, we use a linear extraction mapping $A_x$, and the max player algorithm's output is given by $\mathrm{alg}_{\theta,x}(\cdot \mid D_{x,t-1}) =$

softmax $(A_x \cdot \text{TF}_{x,\theta}(h(D_{x,t-1})))$, where $D_{x,t-1}$ represents the extended history that might include additional context beyond just the max player's own history. In the case of Markov game, the output becomes $\text{alg}_{\theta,x}(\cdot \mid D_{x,t-1}, s_t) = \text{softmax}(A_x \cdot \text{TF}_{x,\theta}(h(D_{x,t-1}, s_t)))$.

**Transformer Components** The Transformer architecture has become a foundational model for sequence modeling tasks due to its ability to capture long-range dependencies using attention mechanisms. We describe the three types of layers relevant to our framework.

1) Self-Attention Layer. For each token embedding $h_i$ in the sequence, the self-attention mechanism computes $\bar{h}_i = h_i + \sum_{m=1}^{M} \sigma\left(\frac{\langle W_m^K h_i, W_m^Q h_i \rangle}{\sqrt{d}}\right) W_m^V h_i$, where $M$ is the number of attention heads, $\sigma$ is the ReLU activation function, $W_m^K, W_m^Q, W_m^V \in \mathbb{R}^{d \times d}$ are the Key, Query, and Value projection matrices for the $m$-th attention head, $\sqrt{d}$ is a scaling factor to stabilize gradients during training.

2) Masked Attention Layer. To enforce causal attention, for each token embedding $h_i$, the masked attention mechanism computes $\bar{h}_i = h_i + \sum_{m=1}^{M} \frac{1}{i} \sum_{j=1}^{i} \sigma\left(\frac{\langle K_m h_i, Q_m h_j \rangle}{\sqrt{d}}\right) \cdot V_m h_j$, where $K_m, Q_m, V_m \in \mathbb{R}^{d \times d}$ are the Key, Query, and Value projection matrices for the $m$-th attention head.

3) MLP Layer. After attention, each token representation is further processed through a position-wise feed-forward network $\bar{h}_i = h_i + W^{L_1} \cdot \sigma(W^{L_2} \cdot h_i)$, where $\sigma$ is the ReLU activation function, $W^{L_2} \in \mathbb{R}^{d \times d'}$ projects to an intermediate dimension $d'$, $W^{L_1} \in \mathbb{R}^{d' \times d}$ projects back to the model dimension $d$.

**Complete $L$-layer $M$-head Transformer** Now we formally define the Transformer that we will investigate in the following definition.

**Definition 3.1 (Decoder-based Transformer Shi et al. (2024))** *An $L$-layer $M$-head decoder-based transformer, denoted as $\text{TF}_{\boldsymbol{\theta}}(\cdot)$, is a composition of $L$ masked or self attention layers, each followed by an MLP layer $\text{TF}_{\boldsymbol{\theta}}(\boldsymbol{H}) = \boldsymbol{H}^{(L)} \in \mathbb{R}^{d \times N}$, where $\boldsymbol{H}^{(L)}$ is defined iteratively by taking $\boldsymbol{H}^{(0)} = \boldsymbol{H} \in \mathbb{R}^{d \times N}$ and for $l \in [L]$, $\boldsymbol{H}^{(l)} = \text{MLP}_{\boldsymbol{\theta}_{\text{mlp}}^{(l)}}\left(\text{Attn}_{\boldsymbol{\theta}_{\text{attn}}^{(l)}}\left(\boldsymbol{H}^{(l-1)}\right)\right) \in \mathbb{R}^{d \times N}$, where $\theta_{\text{attn}}^{(l)} \in \mathbb{R}^{d \times d}$, and $\theta_{\text{mlp}}^{(l)} = (W_l^{L_1}, W_l^{L_2}) \in \mathbb{R}^{d' \times d} \times \mathbb{R}^{d \times d'}$, $\text{Attn}$ refers to the masked attention layer that has $M$ heads.*

The stacking of $L$ such layers allows the transformer to learn increasingly complex patterns and dependencies in the data, with each layer building upon the representations learned by the previous layers.

We define the parameter class of transformers as $\Theta_{d,L,M,d',F} := \left\{ \boldsymbol{\theta} = \left( \boldsymbol{\theta}_{\text{self}}^{(1:L_1)}, \boldsymbol{\theta}_{\text{mattn}}^{(1:L_2)}, \boldsymbol{\theta}_{\text{mlp}}^{(1:L_3)} \right) : \|\boldsymbol{\theta}\| \leq F \right\}$, where $\boldsymbol{\theta}_{\text{self}}^{(1:L_1)}$ represents the parameters of $L_1$ self-attention layers, $\boldsymbol{\theta}_{\text{mattn}}^{(1:L_2)}$ represents the parameters of $L_2$ masked attention layers, $\boldsymbol{\theta}_{\text{mlp}}^{(1:L_3)}$ represents the parameters of $L_3$ MLP layers, and $L_1 + L_2 + L_3 = L$ is the total number of layers in the transformer. Given a set of parameters, the norm $\|\boldsymbol{\theta}\|$ of a Transformer is defined in Appendix F.

# 4 MAIN RESULTS

To demonstrate the in-context game-playing capabilities of transformers, we first establish the result that the transformer family can implement Online Mirror Descent (OMD) algorithm, which is a well-known algorithm proven to achieve last-iterate convergence in (Markov) zero-sum games Cen et al. (2023); Cai et al. (2024). While prior works commonly employ KL-divergence or entropy-based regularizers on OMD, such an implementation might not be feasible for Transformers. Instead, we consider OMD with a squared norm regularizer. The unique challenges associated with realizing OMD under different regularization schemes will be discussed in subsequent sections.

## 4.1 REALIZING OMD WITH TRANSFORMERS

Recall that prior works were able to establish the convergent dynamics for Transformers in zero-sum games Shi et al. (2024), but the Transformer investigated there was a modified version with an

external sampling step. The reliance on an external sampling algorithm stems from limitations in the transformer-realized learning algorithm, V-learning, which lacks last-iterate convergence guarantees. Instead, it depends on an external sampling mechanism to stochastically average across historical iterates, effectively transforming an average-iterate guarantee into a last-iterate guarantee. This requirement for external sampling is not inherent to the in-context capabilities of transformers, but rather an artifact of the implemented algorithm and its corresponding analysis.

To eliminate this artificial sampling layer during inference tasks, we introduce a new approach to the analysis. We demonstrate that the transformer architecture can implement the OMD algorithm with a certain degree of accuracy. OMD has been proven to enjoy last-iterate convergence guarantees across various game settings with a negative entropy regularizer Cai et al. (2024). However, the KL divergence induced by the negative entropy can be non-smooth at the edge of the probability simplex. Because non-smooth operation is generally hard to approximate by sums of ReLUs, it poses a challenge for the transformer to approximate negative entropy regularizers.

It is then intuitive to find naturally smooth regularizers, such as the squared norm. Fortunately, it is easy to extend the convergence guarantee to OMD with a squared norm. In this work, we consider the following Online Mirror Descent (OMD) algorithm with a squared Euclidean norm regularizer, which takes the update form of $x_{t+1} = \arg\min_{x \in \Omega} \left\{ \langle x, g_t \rangle + \frac{1}{\eta} \|x - x_t\|_2^2 \right\}$. where the feasible set is defined as $\Omega = \left\{ x \in \Delta(A) : x_a \geq \frac{1}{AT^2} \right\}$, a truncated simplex that enforces a minimum action probability. The update direction $g_t \in \mathbb{R}^{|A|}$ is given by $g_{t,a} = \frac{r_t \cdot \mathbb{I}\{a_t = a\}}{x_{t,a} + \tilde{\beta}} - \epsilon x_{t,a}$, where $r_t$ is the observed reward, $\mathbb{I}\{a_t = a\}$ is the indicator for the selected action at time $t$, $\tilde{\beta} > 0$ is a constant to encourage exploration, and $\epsilon > 0$ is an regularization term.

A central challenge in realizing this variant of Online Mirror Descent (OMD) with transformers lies in the nature of the update rule. Existing proofs of transformers' capability to realize online algorithms typically depend on the algorithm having either (i) a smooth, closed-form update rule directly approximable by a sum of ReLUs, or (ii) an update procedure that can be emulated through iterative gradient descent steps. In these scenarios, transformers can approximate the update by learning a smooth function directly or by simulating gradient descent iterations. For example, Shi et al. (2024) demonstrates that transformers can efficiently implement a variant of the multiplicative weights algorithm with closed-form, smooth updates, while Bai et al. (2024) shows that transformers can approximate LinUCB by effectively simulating its gradient descent steps.

Neither approach is viable when it comes to proving that transformers can implement OMD with last-iterate convergence guarantees. A critical component of OMD's last-iterate convergence lies in its constraint set—a truncated probability simplex—within the minimization objective. This update lacks a general closed-form expression. Furthermore, since our OMD update requires solving a constrained optimization problem, it cannot be achieved through simple unconstrained gradient steps. Yet projection onto the constrained simplex is a nontrivial operation that is not easily implemented with standard transformer components. To overcome this, we first show that a transformer can approximate an unconstrained update rule of $x_{t+1} = \arg\min_x u(x)$, where the objective $u$ is obtained from the Lagrangian multiplier method. Then we analyze the error given by the approximate solution of $u(x)$, and carefully analyze the smoothness of $u(x)$.

Theorem 4.1 formally characterizes the architectural requirements for a transformer to realize the proposed OMD algorithm. Notably, the dimension of the transformer and embedding needed only scales linearly with the action dimension.

**Theorem 4.1** *For any $\epsilon > 0$, there exists a Transformer $TF$ with $d = O(A)$, $d' = d_D + O(A)$, $L = L_D T$, $\max_{l \in L} M_l = \max\{M_D, O(c(A) \log(1/\epsilon)/\epsilon^2)\}$, and the Transformer norm defined in Appendix F $\|\theta\| = \max\{F_D, 3\}$, such that for all $\bar{D}_{x,t-1}$, $t \in [T]$, $\|\mathsf{Alg}(\cdot \mid \bar{D}_{x,t-1}) - \mathsf{Alg}_{OMD}(\cdot \mid \bar{D}_{x,t-1}))\|_2 \leq \epsilon + \eta A$. A similar construction exists for the y-player.*

*Here, the $d_D, L_D, M_D$ are constants from the following assumption that the transformer can perform exact division, $\mathsf{Alg}_{OMD}$ is the update dynamics of OMD with squared norm.*

**Assumption 4.1 (Assumption G.2 of Shi et al. (2024))** *There exist a Transformer with $L = L_D$, $\max_{l \in L} M_l = M_D$, $d' = d_D$, $\|\theta\| = F_D$ such that for any $x, y \in [0, 1]$, for input $H = [h_1, \ldots]$, where $h_i = [y, x, 1]$, the output is $h_i = [y, x/y, 1]$.*

We remark that the assumption of exact division is only for the simplicity of the analysis and can be removed.

## 4.2 IN-CONTEXT LEARNING CAPABILITY IN ZERO-SUM GAME

Intuitively, the in-context game-playing capability of the transformer depends on several key factors. First, given access to an optimal maximum likelihood estimator, we ask how accurately it can approximate the expert's actions from the provided dataset, which is captured by the notion of approximate realizability of the dataset. Second, the learning difficulty is influenced by the covering number of the parameter class $\Theta$, which quantifies the complexity of identifying the optimal parameter. Lastly, successful in-context learning also hinges on the last-iterate convergence property of the OMD algorithm and the transformer's ability to accurately implement its dynamics. Definition 4.1 and Definition 4.2 formalize the concepts of approximate realizability and the covering number of $\Theta$, respectively.

**Definition 4.1 (Approximate realizability)** *For some $\epsilon_{real} > 0$, if there exist an optimal maximum likelihood estimator $\theta^* \in \Theta$ and $\epsilon_{real} > 0$ such that for all $t \in [T]$, $\mathbb{E}_{G \sim \Lambda, \bar{D}_T \sim \mathbb{P}_G^{Alg_0, Alg_E}} \left[ \mathrm{TV} \left( \overline{\mathrm{Alg}}_E(\cdot \mid D_{t-1}), \mathrm{Alg}_{\theta^*}(\cdot \mid D_{t-1}) \right) \right] \leq \epsilon_{real}$ we say that $\Theta$ is be $\epsilon_{real}$-realizable.*

**Remark 4.1** *We note that previous works measure approximate realizability using log-likelihood Shi et al. (2024); Lin et al. (2024). In contrast, we use the total variation (TV) distance. Bounding the TV distance guarantees that the models behave similarly in every situation, rather than behaving similarly merely on average.*

Having established the capabilities of the optimal parameter, we next define the covering number of the Transformer algorithm class to characterize the complexity of learning such an optimal parameter from finite data.

**Definition 4.2 (Covering number Shi et al. (2024); Lin et al. (2024))** *For a class of Algorithm $\{Alg_\theta, \theta \in \Theta\}$, we say $\Theta_0 \subseteq \Theta$ is an $\rho$-covering of $\Theta$ if $\Theta_0$ is a finite set such that for any $\theta \in \Theta$, there exists some $\theta_0 \in \Theta_0$, $\|\log Alg_{\theta_0}(\cdot \mid D_{t-1}) - \log Alg_\theta(\cdot \mid D_{t-1})\|_\infty \leq \rho$, for all $D_{t-1}, t \in [T]$. The covering number $ncov(\rho)$ is the minimal cardinality of $\Theta_0$ such that $\Theta_0$ is a $\rho$ covering set.*

Our main statement, Theorem 4.2, characterizes the in-context capacity of the transformers in zero-sum games.

**Theorem 4.2** *Let $\Theta_x$ and $\Theta_y$ be the classes of transformers satisfying the requirements in Theorem 4.1 and $Alg_{OMD, \Lambda}$ be the OMD algorithm. Let $x_{\hat{\theta}(D)}, y_{\hat{\theta}(D)}$ be the output policies of the transformer trained on the dataset $D_T$. Then, with probability at least $1 - \delta$, it holds that $\mathbb{E}_{G' \sim \Lambda} \left[ \mathbb{E}_{D \sim P_G^{Alg_0, \Lambda}} \left[ \max_{x,y} \left( x_{\hat{\theta}(D)}^\top A_{G'} y - x^\top A_{G'} y_{\hat{\theta}(D)} \right) \right] \right] \leq O \left( T^{-1/2} \|z^* - z_0\|^2 + AT^{-1/6} + \sqrt{\ln(AT/\delta)} T^{-7/12} + \sqrt{\epsilon_{real} T} + \frac{\log(ncov(1/n)/\delta)}{n} + \epsilon_{real} \right)$, where $ncov(\cdot)$ is the covering number and $z_0 = (x_0, y_0)$ is the initialized policy of the transformer.*

We note that although the result scales with $\epsilon_{approx} T$, but such term $\epsilon_{approx}$ can be controlled to be arbitrarily small by scaling the size of the transformer as described in Theorem 4.1.

The result of Theorem 4.1 can be decomposed into several contributing terms. The first term, $O(T^{-3/8})$, arises from the last-iterate convergence rate of the OMD algorithm in zero-sum games. The second term, $\epsilon_{real} T$, captures the cumulative error incurred by the transformer's approximation of the OMD updates over $T$ iterations. The third term, $\log(ncov(1/n)/\delta)/n$, reflects the statistical complexity of learning the optimal parameter from a dataset of $n$ samples, as governed by the covering number of the parameter space. Finally, the term $\epsilon_{real}$ quantifies the approximate realizability, which is how well the optimal parameter can reproduce the expert's actions.

**Remark 4.2** *In contrast to previous work, we show that Transformers can approximate online mirror descent (OMD) with last-iterate guarantees up to an $\epsilon$ error. This reveals a fundamental tradeoff between approximation accuracy and Transformer size. This tradeoff would not have existed if*

*Transformers realized the multiplicative weights algorithm with an external sampling mechanism. Our results offer new insights into the intrinsic tradeoffs between Transformer architecture design and game-playing capability achieved.*

### 4.3 EXTENSION TO MARKOV ZERO-SUM GAMES

Let $n_{t+1}(s_t)$ be the number of visits to state $s_t$ up to time $t$, then the OMD update rule for the Markov zero-sum game is described in the following pseudocode.

---

**Algorithm 1** OMD for Markov zero-sum game

---

1: **Input:** Parameters $\alpha_\tau, \eta, \epsilon, \beta_\tau$
2: **Initialization:** $x_{1,a}(s) = 1/A$, for all $a$, $n_1(s) = 0$, $V_1(s) = \frac{1}{2(1-\gamma)}$, for all $s$.
3: **for** $t = 1, \ldots, T$ **do**
4:    $\tau \leftarrow n_{t+1}(s_t)$, and Sample $a_t$ from $x_t$.
5:    Set $g_{t,a}(s_t) \leftarrow \left( \frac{\mathbb{I}\{a_t=a\}(r_t+\gamma V_t(s_{t+1}))}{x_{t,a}(s_t)+\beta_\tau} \right) - \epsilon x_{t,a}(s_t)$ for each action $a$.
6:    Set $x_{t+1}(s_t) \leftarrow \arg\min_{x \in \Omega} \langle x, g_t \rangle + \frac{1}{\eta} \|x - x_t(s_t)\|_2$
7:    $V_{t+1}(s_t) \leftarrow (1-\alpha_\tau)V_t(s_t) + \alpha_\tau(r_t + \gamma V_t(s_{t+1}))$
8:    No update for $s \neq s_t$
9: **end for**

---

We note that this algorithm closely resembles the one used to achieve last-iterate convergence in Markov zero-sum games.

Consistent with prior works, our results assume that the Markov game is irreducible and that the travel time between any pair of states is uniformly bounded. This assumption is essential for ensuring efficient exploration and is also required in the strongest known results on last-iterate convergence for Markov zero-sum games Cai et al. (2024).

**Assumption 4.2** *We assume that under any pair of stationary policies of the two players, and any pair of states $s, s'$, the expected time to reach $s'$ from $s$ is upper bounded by $L$.*

Similar to the analysis of zero-sum games, we first show that Transformers with certain architectures can realize OMD.

**Theorem 4.3** *For any $\epsilon > 0$, there exists a transformer $TF$ with $L = (L_D + 1)T$, $\max_{l \in [L]} M^l = O(S^2 + \max\{M_D, O(c(A)\log(1/\epsilon)/\epsilon^2)\})$, $d' = O(T + A)$, the Transformer norm defined in Appendix F $\|\theta\| = O(T^3)$, $d = O(A)$, such that for all $\bar{D}_{x,t-1}$, $t \in [T]$, $\|\mathsf{Alg}(\cdot \mid \bar{D}_{x,t-1}, s_t) - \mathsf{Alg}_{OMD}(\cdot \mid \bar{D}_{x,t-1}, s_t)\| \leq \epsilon + \eta A$. A similar construction exists for the $y$-player.*

The in-context game-playing capacity is formally captured by the following Theorem.

**Theorem 4.4** *Let $\Theta_x$ and $\Theta_y$ be the classes of transformers satisfying the requirements in Theorem 4.1 and $\mathsf{Alg}_{OMD,\Lambda}$ be the OMD algorithm. Let $x_{\hat{\theta}(D)}, y_{\hat{\theta}(D)}$ be the output policies of the transformer trained on the dataset $D_T$. Then, with probability at least $1 - \delta$, it holds that $\mathbb{E}_{G' \sim \Lambda} \left[ \mathbb{E}_{D \sim P_G^{\mathsf{Alg}_0,\Lambda}} \left[ \max_{x,y,s} \left( V_{x_T,y}^s - V_{x,y_T}^s \right) \right] \right] = O\left( \sqrt{L} T^{-1/4} \|z^{*,s} - z_0^s\| + ALT^{-1/6} + \sqrt{\varepsilon_{real} T} + \epsilon_{real} + \frac{\log(\mathrm{ncov}(1/n)/\delta)}{n} + \frac{A \ln^3(SAT/\delta)}{(1-\gamma)^3} \left( \frac{L \ln(ST/\delta) \ln(T/(1-\gamma))}{1-\gamma} \right) \frac{L \ln(ST/\delta)}{T}^{1/3} \right)$, where $z_0^s = (x_0^s, y_0^s)$ is the initialized policy of the transformer.*

Similar to the results in the zero-sum game setting, we note that although the bound scales with $\epsilon_{\mathrm{approx}} T$, the term $\epsilon_{\mathrm{approx}}$ can be made arbitrarily small by appropriately scaling the size of the Transformer, as described in Theorem 4.3.

Inspecting the bound provides us with insights into the efficiency of training Transformers with respect to the length of the trajectories.

**Remark 4.3** *Previous error bound in Shi et al. (2024), which scales linearly with $T$, suggests a less efficient training with longer trajectory (i.e., larger $T$). In contrast to that, our results show that the duality gap in in-context game playing decreases as $T$ increases, provided the Transformer architecture is sufficiently large to ensure a small $\epsilon_{\text{real}}$. Intuitively, a longer trajectory allows the model to learn more effectively from repeated interactions.*

## 5 EXPERIMENTS

Our experimental setup for the zero-sum games involved training transformer models on trajectories of varying lengthsnacross a set of 5 similar zero-sum games with $3, 5$ actions per player. The games were generated by perturbing a base payoff matrix with controlled perturbation magnitude ($\rho = 0.1$). For each trajectory length, we trained separate models for both max and min players using supervised learning on expert demonstrations derived from Nash equilibrium strategies. Models were then evaluated on unseen games by measuring exploitability, which is the maximum advantage an optimal counter-strategy could achieve against the learned policies.

We then extend our experimental results to Markov games, which introduce sequential state transitions and longer-term strategic planning. The similar Markov games, each with $2, 3, 5$ states and $3$ actions per player, where transition probabilities and reward matrices were perturbed from a base game with controlled perturbation magnitude ($\rho = 0.1$). We then train and test the transformer models, similar to what we do with the zero-sum case.

For both zero-sum and Markov zero-sum game results, we use a transformer-based architecture with multi-head attention mechanisms ($4$ heads) and a model dimension of $64$. The model consisted of $2$ transformer layers with a feed-forward dimension of $256$. For input representation, we use a tokenization scheme that encoded states (if any), actions, and discretized rewards into a unified vocabulary space. We trained the models using the Adam optimizer with a learning rate of $0.001$ and employed early stopping when the cross-entropy loss fell below $0.01$.

We repeated each experiment with $5$ different random seeds to ensure statistical robustness, with the mean and standard deviation reported. The following results demonstrate how trajectory length affects the algorithms' ability to generalize to new games within the same distribution with limited size of the transformer model. This validates tradeoff between trajectory length and model performance.

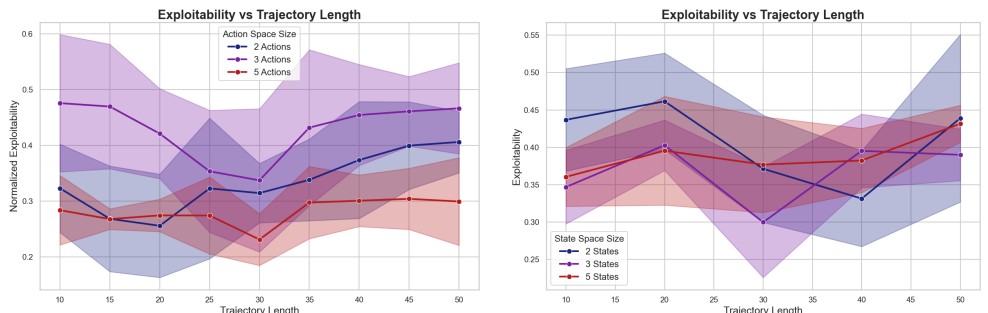

Figure 1: Exploitability on zero-sum game with $[2, 3, 5]$ actions (Left), exploitability on Markov sum game zero-sum game with $[2, 3, 5]$ states and $3$ actions (Right)

## 6 CONCLUSION

This work shows that Transformers, in their standard form, can perform in-context learning in zero-sum and Markov zero-sum games. We establish theoretical bounds on learning accuracy, revealing key architectural factors for effective gameplay and highlighting tradeoffs between model size, strategic precision, and trajectory length. Notably, we provide the first theoretical proof that Transformers can implement no-regret dynamics like online mirror descent in both matrix and Markov games. The finding could be of independent interest in analyses of Transformers in other tasks.

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

## A  PROOF OF THEOREM 4.1

A key tool to our analysis is the approximbility by the sum of ReLUs. Specifically, one can show that if a function is smooth, it is approximable by a sum of ReLUs, which can be efficiently implemented by transformers. The following definitions formally define the approximability and the smoothness of a function.

**Definition A.1 (Approximability by Sum of ReLUs, Definition 12 in Bai et al. (2024))** *A function* $g : \mathbb{R}^k \to \mathbb{R}$ *is* $(\varepsilon_{approx}, R, M, C)$-*approximable by sum of ReLUs, if there exists a "*$(M, C)$-*sum of ReLUs" function* $f_{M,C}(\boldsymbol{z}) = \sum_{m=1}^{M} c_m \sigma \left( \boldsymbol{a}_m^\top [\boldsymbol{z}; 1] \right)$, *with*

$$\sum_{m=1}^{M} |c_m| \le C, \quad \max_{m \in [M]} \|\boldsymbol{a}_m\|_1 \le 1, \quad \boldsymbol{a}_m \in \mathbb{R}^{k+1}, \quad c_m \in \mathbb{R}$$

*such that* $\sup_{\boldsymbol{z} \in \mathrm{Ball}_\infty^k(R)} |g(\boldsymbol{z}) - f_{M,C}(\boldsymbol{z})| \le \varepsilon_{approx}$.

**Definition A.2 (Definition A.1 in Bai et al. (2024))** *We say a function $g : \mathbb{R}^k \to \mathbb{R}$ is $(R, C_\ell)$-smooth if for $s = \lceil (k-1)/2 \rceil + 2$, $g$ is a $C^s$ function on $\mathrm{Ball}_\infty^k(R)$, and*

$$\sup_{\boldsymbol{z} \in \mathrm{Ball}_\infty^k(R)} \left\| \nabla^i g(\boldsymbol{z}) \right\|_\infty = \sup_{\boldsymbol{z} \in \mathrm{Ball}_\infty^k(R)} \max_{j_1, \cdots, j_i \in [k]} \left| \partial_{z_{j_1} \cdots z_{j_i}} g(\boldsymbol{z}) \right| \leq L_i$$

*for all $i \in \{0, 1, \cdots, s\}$, with $\max_{0 \leq i \leq s} L_i R^i \leq C_\ell$.*

**Proposition A.1 (Proposition A.1 in Bai et al. (2024))** *For any $\varepsilon_{approx} > 0, R \geq 1, C_\ell > 0$, we have the following: Any $(R, C_\ell)$ smooth function $g : \mathbb{R}^k \to \mathbb{R}$ is $(\varepsilon_{approx}, R, M, C)$ approximable by sum of ReLUs with $M \leq C(k) C_\ell^2 \log \left( 1 + C_\ell / \varepsilon_{approx} \right) / \varepsilon_{approx}^2$ and $C \leq C(k) C_\ell$, where $C(k) > 0$ is a constant that depends only on $k$.*

One challenge of the proof is that the OMD step is not directly approximated by the sum of ReLUs, making it hard to approximate with a transformer. The following Lemma shows that the implementation of OMD can be done by approximating a surrogate objective.

**Lemma A.1** *There exists a $(M, C)$-sum of ReLUs function $\hat{x}_{t+1} = f_{M,C}(\boldsymbol{z}) = \sum_{m=1}^M c_m \sigma \left( \boldsymbol{a}_m^\top [\boldsymbol{z}; 1] \right)$, with*

$$\sum_{m=1}^M |c_m| \leq C, \quad \max_{m \in [M]} \|\boldsymbol{a}_m\|_1 \leq 1, \quad \boldsymbol{a}_m \in \mathbb{R}^{k+1}, \quad c_m \in \mathbb{R}$$

*such that $\|\hat{x}_{t+1} - x_{t+1}\|_1 \leq \epsilon_{real} + \eta A$, where $M = O(c(A) \log(1/\epsilon_{real})/\epsilon_{real}^2)$, and $C = O(c(A))$, where $c(A)$ is a constant that only depends on $A$.*

**Proof:** We thus would need to reformulate this to $x_{t+1} = \arg\min_x u(x)$, where the minimization is unconstrained and $u$ is smooth. We choose $u(x)$ to be

$$u(x) = \langle x, g_t \rangle + \frac{1}{\eta} \|x - x_t\|^2 + \frac{1}{\lambda} \log \left( \sum_{a=1}^A e^{\lambda(x_a - \frac{1}{AT^2})} \right) + \frac{1}{\lambda} \log \left( e^{\lambda(\sum_{a=1}^A x_a - 1)} + e^{-\lambda(\sum_a x_a - 1)} \right).$$

We first analyze the error given by the approximate solution given by $u(x)$. Let $\tilde{x}_{t+1}$ be the updated policy with $u(x)$, by optimality conditions, we have

$$\langle \nabla u(\tilde{x}_{t+1}), x - \tilde{x}_{t+1} \rangle \geq 0 \quad \langle g_t + \frac{2}{\eta}(x_{t+1} - x_t), x - x_{t+1} \rangle \geq 0.$$

Taking $x = x_{t+1}$ in the first inequality and $x = \tilde{x}_{t+1}$ in the second inequality, and subtracting them gives

$$\langle \nabla u(\tilde{x}_{t+1}) - g_t - \frac{2}{\eta}(x_{t+1} - x_t), x_{t+1} - \tilde{x}_{t+1} \rangle \geq 0.$$

Recall that:

$$\nabla u(\tilde{x}_{t+1}) = g_t + \frac{2}{\eta}(\tilde{x}_{t+1} - x_t) + \nabla \phi(\tilde{x}_{t+1})$$

where $\phi(x) = \frac{1}{\lambda} \log \left( \sum_{a=1}^A e^{\lambda(x_a - \frac{1}{AT^2})} \right) + \frac{1}{\lambda} \log \left( e^{\lambda(\sum_{a=1}^A x_a - 1)} + e^{-\lambda(\sum_a x_a - 1)} \right)$. Substituting this gives

$$\left\langle \frac{2}{\eta}(\tilde{x}_{t+1} - x_{t+1}) + \nabla \phi(\tilde{x}_{t+1}), x_{t+1} - \tilde{x}_{t+1} \right\rangle \geq 0.$$

Therefore,

$$\frac{2}{\eta} \|\tilde{x}_{t+1} - x_{t+1}\|_2^2 \leq \langle \nabla \phi(\tilde{x}_{t+1}), \tilde{x}_{t+1} - x_{t+1} \rangle$$

$$\leq \|\nabla \phi(\tilde{x}_{t+1})\|_2 \|\tilde{x}_{t+1} - x_{t+1}\|_2.$$

For our smooth term, we notice that

$$\partial_j \phi(x) = \frac{e^{\lambda(x_j - \frac{1}{AT^2})}}{\sum_{a=1}^{A} e^{\lambda(x_a - \frac{1}{AT^2})}} + \frac{e^{\lambda(\sum_a x_a - 1)} - e^{-\lambda(\sum_a x_a - 1)}}{e^{\lambda(\sum_a x_a - 1)} + e^{-\lambda(\sum_a x_a - 1)}}$$

$$= \text{softmax}(x_j) + \tanh\left(\lambda(\sum_a x_a - 1)\right).$$

Since both softmax and tanh are bounded by 1, we have $|\partial_j \phi(x)| \leq 2$. Therefore, $\|\nabla \phi(\tilde{x}_{t+1})\|_2 \leq 2\sqrt{A}$. Hence

$$\frac{2}{\eta} \|\tilde{x}_{t+1} - x_{t+1}\|_2^2 \leq 2\sqrt{A} \|\tilde{x}_{t+1} - x_{t+1}\|_2, \tag{3}$$

and $\|\tilde{x}_{t+1} - x_{t+1}\|_2 \leq \sqrt{A}\eta$.

We next show that $u(x)$ is smooth and can be approximated by a sum of ReLUs. For any $x$, the partial derivative of the $j$-th component is

$$\partial_j u(x) = g_{t,j} + \frac{2}{\eta}(x_j - x_j^t) + \frac{e^{\lambda(x_j - \frac{1}{AT^2})}}{\sum_{a=1}^{A} e^{\lambda(x_a - \frac{1}{AT^2})}} + \frac{e^{\lambda(\sum_a x_a - 1)} - e^{-\lambda(\sum_a x_a - 1)}}{e^{\lambda(\sum_a x_a - 1)} + e^{-\lambda(\sum_a x_a - 1)}}.$$

For the first term, $|g_{t,j}| \leq \frac{1}{\beta} + \epsilon \ln(AT^2)$. For the second term, $|\frac{2}{\eta}(x_j - x_j^t)| \leq \frac{2}{\eta}$ for $x \in \text{Ball}_\infty^A(R)$. The second and third terms are bounded by 1 as they are softmax and tanh.

For higher-order derivatives, notice that the first term vanishes after the first order, and the second term vanishes after the second order. For the log-sum-exp terms, all derivatives are combinations of exponential functions divided by sums of exponentials, which are always bounded because: when inputs are large, the exponential terms in the numerator and denominator grow at the same rate, when inputs are small, the terms remain bounded due to the constraints. The fourth term is essentially tanh, which is smooth and has bounded derivatives. Therefore, $u(x)$ satisfies Definition A.2 for some $R, C_\ell$.

Now define $\tilde{u}(z_t) = \text{argmin}_x u([x, z_t])$, where we abuse the notation for $u$ to include the other variables, i.e. $z_t = [x_t, g_t, \eta]$. Then $\tilde{x}_{t+1} = \tilde{u}(x_t)$. To see $\text{argmin}_x u([x, z_t])$ is smooth with respect to $x$, we notice the following. As $u$ is strongly convex and has bounded derivatives, $\nabla^2 u$ is uniformly positive definite and $(\nabla^2 u)^{-1}$ exists and is bounded. Therefore, using the Implicit Function Theorem, the first derivative of $\tilde{u}$ exists and is bounded as the derivatives of $u$ are bounded. Further, the higher order derivatives of $\tilde{u}$ are bounded as it is a composition of derivatives of $u$, which are all bounded. As $g_t$ is only involved in the linear term and $x_t$ is only involved in the quadratic term, $\text{argmin}_x u([x, z_t])$ is smooth with respect to $g_t$ and $x_t$. For any $\eta > 0$, and $z \in \text{Ball}_\infty^{3A+1}$, $\text{argmin}_x u([x, z_t])$ is smooth with respect to $\eta$. Therefore, $\tilde{u}(x)$ satisfies Definition A.2 for some $R, C_\ell$ and is approximable by a sum of ReLUs by Proposition A.1. $\qed$

**Proof:** [Proof of Theorem 4.1] Let $\tilde{a}_t$ be a $A$-dimensional one-hot vector that is 1 at $a_t = a$ and 0 everywhere else, $\tilde{r}_t$, $\tilde{\beta}$ be a $A$-dimensional where each entry is $r_t, \beta$ and $\tilde{\epsilon}, \tilde{\eta}$ similarly. So now the problem is just all left to compute $g_t$. We consider the embedding of

$$h_t = [\tilde{r}_t, \tilde{a}_t, \tilde{\beta}, \tilde{\epsilon}, \tilde{\eta}, x_t, 0_A, 0_A, 0_A, 1_A, x_t]^\top.$$

For the $\ldots$, they are filled with place holders of $A$-dimensional vectors of 1.

Therefore, $h_t$ has a dimension of $O(A)$.

**Step 1** In the first layer, we use the $M = 1$ self-attention with ReLU activation. Construct $W_1^k, W_1^Q$ and $W_1^V$ such that

$$W_1^k h_t = \tilde{r}_t, \quad W_1^Q h_t = \tilde{a}_t, \quad \left[\sigma_r\left(\left\langle W_1^k h_t, W_1^Q h_t \right\rangle\right)\right]_a = \mathbb{I}\{a_t = a\}r_t, \quad W_1^v h_t = [*, 1_A, 0_A, 0_A, 0_A],$$

where $*$ denotes the omitted zero entries. We can construct the matrices explicitly with $W_1^k$ being a block-diagonal matrix. The first $A$ columns of $W_1^k$ form an $A \times A$ identity matrix $I_A$, and the

remaining $11A - A$ columns are all zero. So $W_1^k = \mathbf{diag}(I_A, 0_{A \times 10A})$. We can construct the block matrices $W_1^Q$ and $W_1^V$ similarly. The operator norm of $W_1^Q, W_1^V, W_1^k$ is 1.

The output matrix can just be an identity matrix, so the operator norm is also 1.

Therefore the output $\bar{h}_{t,1} = [\tilde{r}_t, \tilde{a}_t, \tilde{\beta}, \tilde{\epsilon}, \tilde{\eta}, x_t, \tilde{r}_t\tilde{a}_t, 0_A, 0_A, 1_A, x_t]^\top$. The norm of the first self-attention layer is $\{1, 1\} + 1 + 1 = 3$.

**Step 2**  In the second layer, we use a MLP with ReLU activation. Let $W_2^{L_2} = [0, 0, 1, 0, 0, 0, \ldots]$, $W_2^{L_1} = [0, 0, 0, 0, 0, 1, \ldots]$. Therefore, $\bar{h}_{t,2} = [\tilde{r}_t, \tilde{a}_t, \tilde{\beta}, \tilde{\epsilon}, \tilde{\eta}, x_t + \tilde{\beta}, \tilde{r}_t\tilde{a}_t, 0_A, 0_A, 1_A, x_t]^\top$. The norm of the second layer is therefore $1 + 1 = 2$.

**Step 3**  We next compute the division and logarithm

To facilitate the analysis, we assume that transformers exist that can compute the division exactly, with Assumption 4.1, we have

$$\bar{h}_{t,3} = \left[\tilde{r}_t, \tilde{a}_t, \tilde{\beta}, \tilde{\epsilon}, \tilde{\eta}, x_t + \tilde{\beta}, \frac{\tilde{r}_t\tilde{a}_t}{x_t + \tilde{\beta}}, 0_A, 0_A, 1_A, x_t\right]^\top.$$

The norm of this layer is therefore $F_D$.

**Step 4**  Next we use a MLP with ReLU to obtain $\bar{h}_{t,4} = \left[\tilde{r}_t, \tilde{a}_t, \tilde{\beta}, \tilde{\epsilon}, \tilde{\eta}, x_t, \frac{\tilde{r}_t\tilde{a}_t}{x_t + \tilde{\beta}}, 0_A, 0_A, 1_A, x_t\right]^\top$. Similar to step 2, the norm of this layer is 2.

**Step 5**  We use the $M = 1$ self-attention with ReLU activation. Construct $W_5^k, W_5^Q$ and $W_5^V$ such that

$$W_5^k h_t = \epsilon, \quad W_5^Q h_t = x_t, \quad \left[\sigma_r\left(\left\langle W_1^k h_t, W_1^Q h_t\right\rangle\right)\right]_a = \epsilon x_t, \quad W_1^v h_t = [*, -1_A, 0_A, 0_A, 0_A],$$

where $*$ denotes the omitted zero entries. Therefore, we have $\bar{h}_{t,5} = \left[\tilde{r}_t, \tilde{a}_t, \tilde{\beta}, \tilde{\epsilon}, \tilde{\eta}, x_t, g_t, 0_A, 0_A, 1_A, x_t\right]^\top$, where $g_t = \frac{\tilde{r}_t\tilde{a}_t}{x_t + \tilde{\beta}} - \epsilon x_t$.

Similar to step 1, the norm of this layer is 3.

**Step 6**  By proposition A.1, there exist a function $f_6(x_t, \eta, g_t) = \sum_{m=1}^M c_m\sigma(a_m^\top x_t + b_m^\top g_t + d_m^\top \tilde{\eta} + e_m)$, with $\sigma$ being the ReLU activation function and $\sum_{m+1}^M \|c_m\| \leq C, |a_m| + |b_m| + |d_m| + |e_m| \leq 1$, that can approximate $x_{t+1}$ up to $\epsilon_{\text{approx}}$ accuracy. We then design $W_{m,6}^q h_t = [0, 0, 0, 0, \tilde{\eta}, x_t, g_t, 0_A, 0_A, 1_A, x_t]^\top$, $W_{m,6}^k h_t = [0, 0, 0, 0, d_m, a_m, g_m, 0_A, 0_A, e_m, 0_A]^\top$, $W_{m,7}^k h_t = [0, 0, 0, 0, 0, 0, 0, 1_A, 0_A, 0, 0_A]^\top$. Therefore, $\bar{h}_{t,5} = \left[\tilde{r}_t, \tilde{a}_t, \tilde{\beta}, \tilde{\epsilon}, \tilde{\eta}, x_t, g_t, f_6(x_t, \eta, g_t), 0_A, 1_A, x_t\right]^\top$.

As $\sum_{m+1}^M \|c_m\| \leq C, |a_m| + |b_m| + |d_m| + |e_m| \leq 1$, the norm of this layer is also 3.

**Step 7**  Next we use a MLP with Relu to obtain $\bar{h}_{t,7} = \left[\tilde{r}_t, \tilde{a}_t, \tilde{\beta}, \tilde{\epsilon}, \tilde{\eta}, 0_A, \frac{\tilde{r}_t\tilde{a}_t}{x_t + \tilde{\beta}}, f_6(x_t, \eta, g_t), 0_A, 1_A, 0_A\right]^\top$. The norm of this layer is 2.

**Step 8**  Next we use a self Attention with $M = 1$ and Relu to obtain $\bar{h}_{t,7} = \left[\tilde{r}_t, \tilde{a}_t, \tilde{\beta}, \tilde{\epsilon}, \tilde{\eta}, f_6(x_t, \eta, g_t), \frac{\tilde{r}_t\tilde{a}_t}{x_t + \tilde{\beta}}, 0_A, 0_A, 1_A, f_6(x_t, \eta, g_t)\right]^\top$. The query, key, and value matrices $W^Q, W^K, W^V$ along with the output projection matrix $W^O$ are constructed to transform the input. As these matrices are of block-identity form, the norm of this self-attention layer is 3.

Overall, this transformer has dimension $d = O(A)$, $L = L_D$, $\max_{l \in L} M_l = M_D$, and $\|\theta\| = \max\{F_D, 3\}$, and can realize one step update of OMD. The overall update can be stacked by $T$ of such transformer. $\square$

## B    PROOF OF THEOREM 4.2

**Proof:**    [Proof of Theorem 4.2]

We want to upper bound

$$\mathbb{E}_{G'\sim\Lambda}\left[\mathbb{E}_{D\sim P_G^{\mathsf{Alg}_0,\Lambda}}\left[\max_{x,y}\left(x_{\hat{\theta}(D)}^\top A_{G'}y - x^\top A_{G'}y_{\hat{\theta}(D)}\right)\right]\right] .$$

Let $(x_o, y_o)$ be the set policy outputted by OMD. Then, we can decompose it as,

$$\left|\max_{x,y}\left(x_{\hat{\theta}(D)}^\top A_{G'}y - x^\top A_{G'}y_{\hat{\theta}(D)}\right) - \max_{x,y}\left(x_o^\top A_{G'}y - x^\top A_{G'}y_o\right)\right|$$

$$\leq \max_{x,y}\left|\left(x_{\hat{\theta}(D)}^\top A_{G'}y - x^\top A_{G'}y_{\hat{\theta}(D)}\right) - \left(x_o^\top A_{G'}y - x^\top A_{G'}y_o\right)\right|$$

$$\leq \max_{x,y}\left|\left(x_{\hat{\theta}(D)} - x_o\right)^\top A_{G'}y - x^\top A_{G'}\left(y_o - y_{\hat{\theta}(D)}\right)\right|$$

$$\leq \|A_{G'}\|_2\left(\left\|x_{\hat{\theta}(D)} - x_o\right\|_2 + \left\|y_{\hat{\theta}(D)} - y_o\right\|_2\right) ,$$

where the last inequality we used the fact that the rewards are bounded between $0, 1$ so the entries $A_{G'}$ must be bounded between $0, 1$. Therefore,

$$\mathbb{E}_{G'\sim\Lambda}\left[\mathbb{E}_{D\sim P_G^{\mathsf{Alg}_0,\Lambda}}\left[\max_{x,y}\left(x_{\hat{\theta}(D)}^\top A_{G'}y - x^\top A_{G'}y_{\hat{\theta}(D)}\right)\right]\right]$$

$$= \mathbb{E}_{G'\sim\Lambda}\left[\max_{x,y}\left(x_o^\top A_{G'}y - x^\top A_{G'}y_o\right)\right] + \sqrt{2}\mathbb{E}_{G'\sim\Lambda}\left[\mathbb{E}_{D\sim P_G^{\mathsf{Alg}_0,\Lambda}}\left[\left(\mathrm{D}_H\left(x_{\hat{\theta}(D)}\|x_o\right) + \mathrm{D}_H\left(y_{\hat{\theta}(D)}\|y_o\right)\right)\right]\right]$$

$$\leq \mathbb{E}_{G'\sim\Lambda}\left[\max_{x,y}\left(x_o^\top A_{G'}y - x^\top A_{G'}y_o\right)\right] + \sqrt{2}\mathbb{E}_{G'\sim\Lambda}\left[\mathbb{E}_{D\sim P_G^{\mathsf{Alg}_0,\Lambda}}\left[\mathrm{D}_H\left(\mathsf{Alg}(\cdot \mid D)\|\mathsf{Alg}_{\mathrm{OMD}}(\cdot \mid D))\right]\right] .$$

Then, combining the results from Lemma B.2 and Lemma B.1 gives the result.    $\square$

**Lemma B.1** *If Definition 4.1 is satisfied, the solution given by Eq. 1 satisfies*

$$\mathbb{E}_{\bar{D}_T}\left[\mathrm{D}_H\left(\mathsf{Alg}(\cdot \mid D_{T-1})\|\mathsf{Alg}_{OMD}(\cdot \mid D_{T-1}))\right] \leq O\left(\frac{\log(\mathrm{ncov}/\delta)}{n}\right) + \epsilon_{\mathrm{real}} ,$$

*with a probability of at least $1 - \delta$.*

**Proof:**    Let $\Theta_0$ be a $1/n$-covering set of $\Theta$ with a covering number of ncov. Then for $k \in [\mathrm{ncov}]$, define $\ell_k^i = \log\left(\frac{\mathsf{Alg}_E(\bar{a}_T^i|\bar{D}_{T-1}^i)}{\mathsf{Alg}_{\theta_k}(\bar{a}_T^i|\bar{D}_{T-1}^i)}\right)$, where $\theta_k$ is some parameter in $\Theta_0$.

Using Lemma E.1 with $X_i = -\ell_k^i/2$ and an union bound gives

$$\sum_{i=1}^n \frac{\ell_k^i}{2} + \log\left(\frac{\mathrm{ncov}}{\delta}\right) \geq \sum_{i=1}^n -\log\mathbb{E}\left[\exp\left(\frac{-\ell_k^i}{2}\right)\right] ,$$

for all $k \in [\mathrm{ncov}]$, with probability of at least $1 - \delta$.

As $\bar{a}_T^i$ are generated by the expert algorithm, we have

$$\mathbb{E}\left[\exp\left(\frac{-\ell_k^i}{2} \mid D_{T-1}^i\right)\right] = \mathbb{E}\left[\sqrt{\mathsf{Alg}_E(\bar{a}_T^i \mid \bar{D}_{T-1}^i)}\mathsf{Alg}_{\theta_k}(\bar{a}_T^i \mid \bar{D}_{T-1}^i)\right]$$

$$= \sum_a \sqrt{\mathsf{Alg}_{\theta_k}(a \mid \bar{D}_{T-1}^i)\mathsf{Alg}_E(a \mid \bar{D}_{T-1}^i)} .$$

Therefore, for any $\theta \in \Theta$ that is covered by $\theta_k$, we have

$$-\log\mathbb{E}\left[\exp\left(\frac{-\ell_k^i}{2}\right)\right]$$

$$\geq 1 - \mathbb{E}_{D_T^i}\left[\sum_a \sqrt{\mathsf{Alg}_{\theta_k}(a \mid \bar{D}_{T-1}^i)\mathsf{Alg}_E(a \mid \bar{D}_{T-1}^i)}\right]$$

$$= 1 - \mathbb{E}_{D_T^i}\left[\sum_a \sqrt{\mathsf{Alg}_\theta(a \mid \bar{D}_{T-1}^i)\mathsf{Alg}_E(a \mid \bar{D}_{T-1}^i)}\right]$$

$$- \mathbb{E}_{D_T^i}\left[\sum_a \sqrt{\mathsf{Alg}_E(a \mid \bar{D}_{T-1}^i)}\left(\sqrt{\mathsf{Alg}_{\theta_k}(a \mid \bar{D}_{T-1}^i)} - \sqrt{\mathsf{Alg}_\theta(a \mid \bar{D}_{T-1}^i)}\right)\right]$$

$$\geq \frac{1}{2}\mathbb{E}_{D_T^i}\left[D_H\left(\mathsf{Alg}_E(\cdot \mid \bar{D}_{T-1}^i)\|\mathsf{Alg}_\theta(\cdot \mid \bar{D}_{T-1}^i)\right)\right] - \left\|\mathsf{Alg}_{\theta_k}(\cdot \mid \bar{D}_{T-1}^i) - \mathsf{Alg}_\theta(\cdot \mid \bar{D}_{T-1}^i)\right\|_1^{1/2}$$

$$\geq \frac{1}{2}\mathbb{E}_{D_T^i}\left[D_H\left(\mathsf{Alg}_E(\cdot \mid \bar{D}_{T-1}^i)\|\mathsf{Alg}_\theta(\cdot \mid \bar{D}_{T-1}^i)\right)\right] - \frac{\sqrt{2}}{n}\,,$$

for all $i \in [n]$, where the first inequality uses $-\log(x) \geq 1 - x$, the second inequality follows the Cauchy-Schwarz inequality, the third inequality uses $(\sqrt{x} - \sqrt{y})^2 \leq |x - y|$ for $x, y \geq 0$ and the last inequality follows from Definition 4.2 and Lemma 15 of Lin et al. (2024).

As any $\theta$ is covered by some $\theta_k$, we have

$$\sum_{i=1}^n \ell_k^i = \sum_{i=1}^n \left(\log \mathsf{Alg}_E(\bar{a}_T^i \mid \bar{D}_{T-1}^i) - \log \mathsf{Alg}_{\theta_k}(\bar{a}_T^i \mid \bar{D}_{T-1}^i)\right)$$

$$\leq \sum_{i=1}^n \left(\log \mathsf{Alg}_E(\bar{a}_T^i \mid \bar{D}_{T-1}^i) - \log \mathsf{Alg}_\theta(\bar{a}_T^i \mid \bar{D}_{T-1}^i) + \frac{1}{n}\right)$$

$$\leq \sum_{i=1}^n \left(\log \mathsf{Alg}_E(\bar{a}_T^i \mid \bar{D}_{T-1}^i) - \log \mathsf{Alg}_\theta(\bar{a}_T^i \mid \bar{D}_{T-1}^i)\right) + 1$$

Therefore,

$$\frac{1}{2}\sum_{i=1}^n \left(\log \mathsf{Alg}_E(\bar{a}_T^i \mid \bar{D}_{T-1}^i) - \log \mathsf{Alg}_\theta(\bar{a}_T^i \mid \bar{D}_{T-1}^i)\right) + \frac{1}{2} + \log\left(\frac{\mathrm{ncov}}{\delta}\right)$$

$$\geq \frac{n}{2}\mathbb{E}_{D_T}\left[D_H\left(\mathsf{Alg}_E(\cdot \mid \bar{D}_{T-1})\|\mathsf{Alg}_\theta(\cdot \mid \bar{D}_{T-1})\right)\right] - \sqrt{2}\,.$$

For the left-hand side, assume $\theta^*$ is the optimal maximum likelihood estimator, then

$$\sum_{i=1}^n \left(\log \mathsf{Alg}_E(\bar{a}_T^i \mid \bar{D}_{T-1}^i) - \log \mathsf{Alg}_\theta(\bar{a}_T^i \mid \bar{D}_{T-1}^i)\right)$$

$$\leq \sum_{i=1}^n \left(\log \mathsf{Alg}_E(\bar{a}_T^i \mid \bar{D}_{T-1}^i) - \log \mathsf{Alg}_{\theta^*}(\bar{a}_T^i \mid \bar{D}_{T-1}^i)\right)$$

$$\leq \sum_{i=1}^n \log \mathbb{E}_{\bar{D}_T}\left[\exp\left(\log \mathsf{Alg}_E(\bar{a}_T^i \mid \bar{D}_{T-1}^i) - \log \mathsf{Alg}_{\theta^*}(\bar{a}_T^i \mid \bar{D}_{T-1}^i)\right)\right] + \log(1/\delta)$$

$$\leq n\epsilon_{\mathrm{real}} + \log(1/\delta)\,.$$

Rearranging this, we have

$$\mathbb{E}_{D_T}\left[D_H\left(\mathsf{Alg}_E(\cdot \mid \bar{D}_{T-1})\|\mathsf{Alg}_\theta(\cdot \mid \bar{D}_{T-1})\right)\right] \leq \frac{1 + 2\sqrt{2} + 2\log(\mathrm{ncov}/\delta)}{n} + \epsilon_{\mathrm{real}}$$

$\square$

**Lemma B.2** *Let $\{x_{o,t}\}_{t=1}^T$ be a sequence of policy that is solved with inaccurate OMD update, i.e.*
$x_{o,t+1} = \hat{x}_{o,t+1} + c_{t+1}\,, \hat{x}_{t+1} = \operatorname{argmin}_{x \in \Omega}\langle x, g_t\rangle + \frac{1}{\eta}\|x - x_t\|_2$, $\Omega = \{x \in \Delta(A) : x_a \geq \frac{1}{AT^2}\}$,
*and* $g_{t,a} = \left(\frac{\mathbb{I}\{a_t = a\}r_t}{x_{t,a} + \beta}\right) - \epsilon x_{t,a}$. *Then,*

$$\max_{x,y}\left(x_{o,T}^\top A_{G'}y - x^\top A_{G'}y_{o,T}\right)$$

$$\leq O\left(T^{-1/2}\|z^* - z_0\|^2 + AT^{-1/6} + \sqrt{\ln(AT/\delta)}T^{-7/12} + \sqrt{\epsilon_{\mathrm{approx}}T}\right)\,.$$

**Proof:** Define $f(x, y) = x^\top G y + \epsilon/2\|x\|_2^2 - \epsilon/2\|y\|_2^2$, and $(x^*, y^*)$ be the equilibrium of $f(x, y)$. Then

$$f(x_t, y_t) - f(x^*, y_t)$$

$$= (x_t - x^*)^\top G y_t + \epsilon/2\|x_t\|_2^2 - \epsilon/2\|x^*\|_2^2$$

$$= (x_t - x^*)^\top g_t + (x_t)^\top (G y_t - g_t) + (x^*)^\top (g_t - G y_t) - \frac{\epsilon}{2}\|x_t - x^*\|_2^2 + \epsilon(x^* - x_t)^\top x_t$$

$$= (x_t - x^*)^\top g_t - \frac{\epsilon}{2}\|x_t - x^*\|_2^2 + \sum_a x_{t,a} \left( (G y_t)_a - \frac{\mathbb{I}\{a_t = a\} r_t}{x_{t,a} + \beta} \right) + \sum_a x_{t,a}^* \left( \frac{\mathbb{I}\{a_t = a\} r_t}{x_{t,a} + \beta} - (G y_t)_a \right).$$

$$(4)$$

By the optimality condition, we have

$$\langle \hat{x}_{t+1} - x^*, \eta g_t + \hat{x}_{t+1} - x_t \rangle \leq 0.$$

Expanding the terms out, we have

$$\langle \hat{x}_{t+1} - x^*, \hat{x}_{t+1} - x_t \rangle + \eta \langle \hat{x}_{t+1} - x^*, g_t \rangle \leq 0.$$

Using the identity $\|x^* - x_t\|^2 = \|x^* - \hat{x}_{t+1}\|^2 + \|\hat{x}_{t+1} - x_t\|^2 + 2\langle \hat{x}_{t+1} - x_t, x^* - \hat{x}_{t+1} \rangle$, we obtain

$$\langle \hat{x}_{t+1} - x^*, g_t \rangle \leq \frac{1}{2\eta} \left( \|x^* - x_t\|^2 - \|x^* - \hat{x}_{t+1}\|^2 - \|\hat{x}_{t+1} - x_t\|^2 \right).$$

Since $\langle x_t - \hat{x}_{t+1}, g_t \rangle \leq \|g_t\|\|x_t - \hat{x}_{t+1}\| \leq \frac{\|x_t - \hat{x}_{t+1}\|^2}{2\eta} + 2\eta\|g_t\|^2$. Hence,

$$(x_t - x^*)^\top g_t \leq \frac{1}{2\eta} \left( \|x^* - x_t\|^2 - \|x^* - \hat{x}_{t+1}\|^2 \right) + 2\eta\|g_t\|^2.$$

Taking this into Equation 4, we have

$$f(x_t, y_t) - f(x^*, y_t) \leq \frac{1 - \eta\epsilon}{2\eta}\|x^* - x_t\|^2 - \frac{1}{2\eta}\|x^* - \hat{x}_{t+1}\|^2 + 2\eta\|g_t\|^2 + \bar{\zeta}_t + \bar{\varrho}_t,$$

where $\bar{\zeta}_t = \sum_a x_{t,a} \left( (G y_t)_a - \frac{\mathbb{I}\{a_t = a\} r_t}{x_{t,a} + \beta} \right)$, $\bar{\varrho}_t = \sum_a x_{t,a}^* \left( \frac{\mathbb{I}\{a_t = a\} r_t}{x_{t,a} + \beta} - (G y_t)_a \right)$.

Rearranging this, we have

$$\|x^* - x_{t+1}\|^2 \leq (1 - \eta\epsilon)\|x^* - x_t\|^2 + 4\eta^2\|g_t\|^2 + 2\eta\bar{\zeta}_t + 2\eta\bar{\varrho}_t + 2\eta f(x^*, y_t) - 2\eta f(x_t, y_t) + \bar{v}_t,$$

where $\bar{v}_t = \|x^* - x_{t+1}\|^2 - \|x^* - \hat{x}_{t+1}\|^2$.

Repeating the same process with $y$ and let $z = (x, y)$, $v_t = (\bar{v}_t, \underline{v}_t)$, $\zeta_t = (\bar{\zeta}_t, \underline{\zeta}_t)$, $\varrho_t = (\bar{\varrho}_t, \underline{\varrho}_t)$, with the underline counterpart be correspond to that for the $y$-player, we obtain

$$\|z^* - z_{t+1}\|^2 \leq (1 - \eta\epsilon)\|z^* - z_t\|^2 + 4\eta^2\|g_t\|^2 + 2\eta\zeta_t + 2\eta\varrho_t + v_t.$$

For any $v_t$, we can decompose it as

$$\|z^* - z_{t+1}\|^2 - \|z^* - \hat{z}_{t+1}\|^2 \leq \|z_{t+1} - \hat{z}_{t+1}\|^2$$
$$\leq \varepsilon_{\text{approx}} + \eta A.$$

Expanding the recursion and letting $w_t = (1 - \eta\epsilon)^t$,

$$\|z^* - z_{t+1}\|^2 \leq (1 - \eta\epsilon)^T \|z^* - z_0\|^2 + 4\sum_{i=1}^t w_t \eta\|g_i\|^2 + 2\sum_{i=1}^t w_t \eta\zeta_i + 2\sum_{i=1}^t w_t \eta\varrho_i + \epsilon_{\text{approx}} T + \eta A T.$$

By Lemma 6 and 7 of Cai et al. (2024), with a probability of at least $1 - \delta/t^2$ we have

$$2\sum_{i=1}^t w_t \eta\zeta_i + 2\sum_{i=1}^t w_t \eta\varrho_i = \eta \cdot O\left( \beta A T + \sqrt{\ln(A T/\delta) T} + \frac{\ln(1/\delta)}{\beta} \right).$$

For the second term, we have $\sum_{i=1}^{t} w_t \eta \|g_i\|^2 \leq O\left(AT^{1+\bar{\beta}-\bar{\eta}}\right)$.

Combining the terms and take $\eta = T^{-\bar{\eta}}$, $\epsilon = T^{-\bar{\epsilon}}$, and $\beta = T^{-\bar{\beta}}$, we have

$$\|z^* - z_{t+1}\|^2$$

$$\leq T^{1-\bar{\eta}-\bar{\epsilon}}\|z^* - z_0\|^2 + O\left(AT^{1+\bar{\beta}-\bar{\eta}} + AT^{1-\bar{\eta}-\bar{\beta}} + \sqrt{\ln(AT/\delta)}T^{1/2-\bar{\eta}} + \ln(1/\delta)T^{\bar{\beta}-\bar{\eta}}\right) + AT^{1-\bar{\eta}} + \epsilon_{\text{approx}}T.$$

Therefore, taking square roots on both sides yields,

$$\max_{x,y}\left(x_{o,T}^\top A_{G'}y - x^\top A_{G'}y_{o,T}\right)$$

$$\leq O\left(T^{-\bar{\epsilon}}\ln(A) + T^{\frac{1-\bar{\eta}-\bar{\epsilon}}{2}}\|z^* - z_0\|^2 + AT^{\frac{1+\bar{\beta}-\bar{\eta}}{2}} + AT^{\frac{1-\bar{\eta}-\bar{\beta}}{2}} + \sqrt{\ln(AT/\delta)}T^{\frac{1/2-\bar{\eta}}{2}} + \ln(1/\delta)T^{\frac{\bar{\beta}-\bar{\eta}}{2}}\right.$$

$$\left. + AT^{\frac{1-\bar{\eta}}{2}} + \sqrt{\epsilon_{\text{approx}}T}\right).$$

Take $\bar{\epsilon} = 1/3$, $\bar{\eta} = 5/3$, $\bar{\beta} = 1/3$, we have

$$\max_{x,y}\left(x_{o,T}^\top A_{G'}y - x^\top A_{G'}y_{o,T}\right)$$

$$\leq O\left(T^{-1/3}\ln(A) + T^{-1/2}\|z^* - z_0\|^2 + AT^{-1/6} + AT^{-1/2} + \sqrt{\ln(AT/\delta)}T^{-7/12} + \ln(1/\delta)T^{-2/3}\right.$$

$$\left. + AT^{-1/3} + \sqrt{\epsilon_{\text{approx}}T}\right)$$

$$\leq O\left(T^{-1/2}\|z^* - z_0\|^2 + AT^{-1/6} + \sqrt{\ln(AT/\delta)}T^{-7/12} + \sqrt{\epsilon_{\text{approx}}T}\right).$$

$\qquad\qquad\qquad\qquad\qquad\qquad\qquad\qquad\qquad\qquad\qquad\qquad\qquad\qquad\qquad\qquad\qquad\qquad\qquad\qquad\qquad\square$

## C   PROOF OF THEOREM 4.4

**Proposition C.1 (Proposition 1 and of Cai et al. (2024))** *If Assumption 4.2 holds, then for any $L' = 2L \log 2(S/\delta)$ consecutive steps, under any (non-stationary) policies of the two players, with probability at least $1 - \delta$, every state is visited at least once. With probability $1 - \delta$, for any $t \geq 1$, players visit every state at least once in every $6L \ln(St/\delta)$ consecutive iterations before time t.*

**Lemma C.1 (Lemma 20 Wei et al. (2021))** *With probability at least $1 - \mathcal{O}(\delta)$, for any state $s \in \mathcal{S}$ and time $t \geq 1$, we have*

$$
|V_t^s - V_\star^s| \leq \mathcal{O}\left( \frac{A \ln(SAt/\delta)}{(1-\gamma)^2} \left( \frac{L \ln(St/\delta)}{1-\gamma} \ln \frac{t}{1-\gamma} \right)^{\frac{k_*}{1-\bar{\alpha}}} \left( \frac{L \ln(St/\delta)}{t} \right)^{k_*} \right)
$$

*where $k_* = \min \left\{ \bar{\eta}, \bar{\beta}, \bar{\alpha} - \bar{\beta}, \bar{\epsilon} \right\}$.*

**Proof:**   [Proof of Theorem 4.4] Similar to that of Theorem 4.2, let $x_o$ be the set of policy outputted by OMD over $T$ iterations, then

$$
\mathbb{E}_{G' \sim \Lambda} \left[ \mathbb{E}_{D \sim P_G^{\mathsf{Alg}_0, \Lambda}} \left[ \max_{x,y,s} \left( V_{x_T, y}^s - V_{x, y_T}^s \right) \right] \right]
$$

$$
= \mathbb{E}_{G' \sim \Lambda} \left[ \max_{x,y,s} \left( V_{x_o, y}^s - V_{x, y_o}^s \right) \right] + \sqrt{2} \mathbb{E}_{G' \sim \Lambda} \left[ \mathbb{E}_{D \sim P_G^{\mathsf{Alg}_0, \Lambda}} \left[ D_H \left( \mathsf{Alg}(\cdot \mid D) \| \mathsf{Alg}_{\mathrm{OMD}}(\cdot \mid D) \right) \right] \right] .
$$

With Lemma B.1, we have the second term be upper bounded as $O\left( \frac{\log(\mathrm{ncov}/\delta)}{n} \right) + \epsilon_{\mathrm{real}}$.

For the first term, by Lemma 10 in Cai et al. (2024) and Wei et al. (2021), we have

$$
\max_{s,x,y} \left( V_{x_t, y}^s - V_{x, y_t}^s \right)
$$

$$
\leq \frac{2}{1-\gamma} \max_{s,x,y} \left( x_t^s \left( G^s + \gamma \mathbb{E}_{s'}[V_t^{s'}] \right) y^s - x^s \left( G^s + \gamma \mathbb{E}_{s'}[V_t^{s'}] \right) y_t^s \right) + \frac{4\gamma}{1-\gamma} \max_s |V_t^s - V_*^s| .
$$

With probability at least $1 - \mathcal{O}(\delta)$, for any $s, x^s, y^s$, and $t \geq 1$, denote $\tau$ the number of visitation to state $s$ until time $t$, then

$$
x_t^s \left( G^s + \gamma \mathbb{E}_{s'} \left[ V_t^{s'} \right] \right) y^s - x^s \left( G^s + \gamma \mathbb{E}_{s'} \left[ V_t^{s'} \right] \right) y_t^s
$$

$$
\leq x_\tau^s \left( G^s + \gamma \mathbb{E}_{s'} \left[ V_{t\tau(s)}^{s'} \right] \right) y^s - x^s \left( G^s + \gamma \mathbb{E}_{s'}^s \left[ V_{t_\tau(s)}^{s'} \right] \right) y_\tau^s + 2 \max_{s'} \left| V_t^{s'} - V_{t_\tau(s)}^{s'} \right|
$$

$$
\leq x^{*,s} \left( G^s + \gamma \mathbb{E}_{s'} \left[ V_{t_\tau(s)}^{s'} \right] \right) y^s - x^s \left( G^s + \gamma \mathbb{E}_{s'} \left[ V_{t_\tau(s)}^{s'} \right] \right) y^{*,s} + 2 \max_{s'} \left| V_t^{s'} - V_{t_\tau(s)}^{s'} \right| + \mathcal{O} \left( \| z_\tau^s - z^{*,s} \|_2 \right)
$$

$$
\leq O(T^{-\epsilon}) + 2 \max_{s'} \left| V_t^{s'} - V_{t_\tau(s)}^{s'} \right| + \mathcal{O} \left( \| z_\tau^s - z^{*,s} \|_2 \right) .
$$

For the last term, define $f(x, y) = x^\top Q_* y + \epsilon/2 \|x\|_2^2 - \epsilon/2 \|y\|_2^2$, and $(x^*, y^*)$ be the equilibrium of $f(x, y)$.

$$
f^s(x_\tau^s, y_\tau^s) - f^s(x^{*,s}, y_\tau^s) \leq \frac{1 - \eta\epsilon}{2\eta} \|x^{*,s} - x_\tau^s\|^2 - \frac{1}{2\eta} \|x^{*,s} - \hat{x}_{\tau+1}^s\|^2 + 2\eta \|g_\tau\|^2 + \bar{\zeta}_\tau + \bar{\varrho}_\tau ,
$$

where

$$
\bar{\zeta}_\tau^s = x_\tau^{s,\top} \left( \left( G^s + \gamma \mathbb{E}_{s'}[V_t^{s'}] \right) y_\tau^s - \frac{\mathbb{I}\{a_\tau^s = a\}(r_\tau + \gamma V_t(s_{t+1}))}{x_{t,a}^s + \beta} \right)
$$

$$
\bar{\varrho}_\tau^s = x_\tau^{s,\top} \left( \frac{\mathbb{I}\{a_\tau^s = a\}(r_t + \gamma V_t(s_{t+1}))}{x_{-,a}^s + \beta} - \left( G^s + \gamma \mathbb{E}_{s'}[V_t^{s'}] \right) y_\tau^s \right) .
$$

Rearranging this, for all $i \leq \tau$, we have

$$
\|x^{*,s} - x_{i+1}^s\|^2 \leq (1 - \eta\epsilon)\|x^{*,s} - x_i^s\|^2 + 4\eta^2 \|g_i^s\|^2 + 2\eta\bar{\zeta}_i^s + 2\eta\bar{\varrho}_i^s + 2\eta f(x^{*,s}, y_i^s) - 2\eta f(x_i^s, y_i^s) + \bar{v}_i^s ,
$$

where $\bar{v}_i^s = \|x^{*,s} - x_{i+1}^s\|^2 - \|x^{*,s} - \hat{x}_{i+1}^s\|^2$.

Repeating the same process with $y$ and let $z = (x, y)$, $v_i^s = (\bar{v}_i^s, \underline{v}_i^s)$, $\zeta_i^s = (\bar{\zeta}_i^s, \underline{\zeta}_i^s)$, $\varrho_i^s = (\bar{\varrho}_i^s, \underline{\varrho}_i^s)$, with the underline counterpart be correspond to that for the $y$-player, we obtain

$$\|z^{*,s} - z_{i+1}^s\|^2 \le (1 - \eta\epsilon)\|z^{*,s} - z_i^s\|^2 + 4\eta^2\|g_i^s\|^2 + 2\eta\zeta_i^s + 2\eta\varrho_i^s + v_i^s \,.$$

Expanding the recursion and letting $w_\tau = (1 - \eta\epsilon)^\tau$,

$$\|z^{*,s} - z_{i+1}^s\|^2 \le (1 - \eta\epsilon)^\tau\|z^{*,s} - z_0^s\|^2 + 4\sum_{i=1}^{\tau} w_\tau\eta\|g_i^s\|^2 + 2\sum_{i=1}^{\tau} w_\tau\eta\zeta_i^s + 2\sum_{i=1}^{\tau} w_\tau\eta\varrho_i^s + \sum_{i=1}^{\tau} w_\tau v_i^s \,.$$

By Lemma 6 and 7 of Cai et al. (2024) and an union bound over $A, S, T$, with a probability of at least $1 - \delta/\tau^2$ we have

$$2\sum_{i=1}^{\tau} w_t\eta\zeta_i^s + 2\sum_{i=1}^{\tau} w_t\eta\varrho_i^s = \eta \cdot O\left(\beta A\tau + \sqrt{\ln(AST/\delta)\tau} + \max_{i\in[\tau]} \frac{\ln(AST/\delta)}{\beta_i}\right) \,.$$

For the second term, we have $\sum_{i=1}^{\tau} w_t\eta\|g_i^s\|^2 \le O\left(A\tau^{1+\bar{\beta}-\bar{\eta}}\right)$. For any $v_t$, we can decompose it as

$$\|z^{*,s} - z_{i+1}^s\|^2 - \|z^{*,s} - \hat{s}_{i+1}^s\|^2 \le \|z_{i+1}^s - \hat{z}_{i+1}^s\|^2$$
$$\le \varepsilon_{\text{approx}} + \eta A \,.$$

Therefore,

$$\|z^{*,s} - z_{i+1}^s\|^2$$
$$\le O\left(\tau^{1-\bar{\eta}-\bar{\epsilon}}\|z^{*,s} - z_0^s\|^2 + A\tau^{1+\bar{\beta}-\bar{\eta}} + A\tau^{1-\bar{\eta}-\bar{\beta}} + \sqrt{\ln(AST/\delta)}\tau^{1-\bar{\eta}} + \ln(AS/\delta)\tau^{\bar{\beta}-\bar{\eta}}\right) + \varepsilon_{\text{approx}}\,\tau + A\tau^{1-\bar{\eta}} \,.$$

Taking $\bar{\epsilon} = 1/3$, $\bar{\eta} = 5/3$, $\bar{\beta} = 1/3$, we have

$$\|z^{*,s} - z_{i+1}^s\|^2$$
$$\le O\left(\tau^{-1}\|z^{*,s} - z_0^s\|^2 + A\tau^{-1/3} + A\tau^{-1} + \sqrt{\ln(AT/\delta)}\tau^{-2/3} + \ln(1/\delta)\tau^{-4/3}\right) + \varepsilon_{\text{approx}}\,\tau + A\tau^{-2/3}$$
$$\le O\left(\tau^{-1}\|z^{*,s} - z_0^s\|^2 + A\tau^{-1/3} + \sqrt{\ln(AST/\delta)}\tau^{-2/3} + \varepsilon_{\text{approx}}\,\tau + A\tau^{-2/3}\right) \,.$$

By Lemma 20 of Cai et al. (2024), we have

$$\frac{4\gamma}{1-\gamma}\max_s|V_t^s - V_*^s| \le \quad O\left(\frac{A\ln^3(SAT/\delta)}{(1-\gamma)^3}\left(\frac{L\ln(ST/\delta)\ln(T/(1-\gamma))}{1-\gamma}\right)^{k_*/(1-\bar{\alpha})}\frac{L\ln(ST/\delta)^{k_*}}{T}\right) \,,$$

where $k_* = \min\{\bar{\alpha} - 1/3, 1/3\}$. Take $\bar{\alpha} = 2/3$, we have

$$\frac{4\gamma}{1-\gamma}\max_s|V_t^s - V_*^s| \le \quad O\left(\frac{A\ln^3(SAT/\delta)}{(1-\gamma)^3}\left(\frac{L\ln(ST/\delta)\ln(T/(1-\gamma))}{1-\gamma}\right)\frac{L\ln(ST/\delta)^{1/3}}{T}\right) \,.$$

Combining everything, for $\tau \ge T/L, \tau \le T$ and have $\varepsilon_{\text{approx}}$ be bounded by $\varepsilon_{\text{real}}$, we have

$$\mathbb{E}_{G'\sim\Lambda}\left[\mathbb{E}_{D\sim P_G^{\text{Alg}_0,\Lambda}}\left[\max_{x,y,s}\left(V_{x_T,y}^s - V_{x,y_T}^s\right)\right]\right]$$
$$= O\left(\sqrt{L}T^{-1/4}\|z^{*,s} - z_0^s\| + ALT^{-1/6} + \sqrt{\ln(AST/\delta)}LT^{-1/3} + \sqrt{\varepsilon_{\text{real}}T} + ALT^{-1/3}\right)$$
$$+ O\left(\frac{A\ln^3(SAT/\delta)}{(1-\gamma)^3}\left(\frac{L\ln(ST/\delta)\ln(T/(1-\gamma))}{1-\gamma}\right)\frac{L\ln(ST/\delta)^{1/3}}{T}\right)$$
$$+ O\left(\frac{\log(\text{ncov}/\delta)}{n}\right) + \epsilon_{\text{real}} \,.$$

$\square$

## D PROOF OF THEOREM 4.3

**Proof:** [Proof of Theorem 4.3] By step 1 and 2 of Theorem 3.4 of Shi et al. (2024), we can realize one-step $V_{t+1}(s_t)$ update with a transformer of size $L = 1$, $\max_{l \in [L]} M^l = O(S^2)$, $d' = O(T)$, $\|\theta\| = O(T^3)$.

By Theorem 4.1, we can stack this and compute $x_{t+1}(s_t)$ with another transformer of size $L = L_D$, $d = O(A)$, $d' = d_D + O(A)$, $\max_{l \in [L]} M^l = M_D$.

Therefore, the overall size of the transformer needed for one-step update is $L = L_D + 1$, $\max_{l \in [L]} M^l = O(S^2 + M_D)$, $d' = O(T + A)$, $\|\theta\| = O(T^3)$, $d = O(A)$.

To perform the overall update, we stack $T$ of such transformers and the overall norm of the transformer is $L = (L_D + 1)T$, $\max_{l \in [L]} M^l = O(S^2 + M_D)$, $d' = O(T + A)$, $\|\theta\| = O(T^3)$, $d = O(A)$. $\square$

## E AUXILIARY LEMMA

**Lemma E.1 (Lemma 14 Lin et al. (2024))** *For any sequence of random variable $\{X_i\}_{i \leq n}$ adapted to a filtration $\{\mathcal{F}_i\}_{i \leq n}$, with probability of at least $1 - \delta$, we have*

$$\sum_{i=1}^{m} X_i \leq \sum_{i=1}^{m} \log \mathbb{E}\left[\exp(X_i) \mid \mathcal{F}_{i-1}\right] + \log(1/\delta), \forall s \in [n].$$

## F NORM OF TRANSFORMERS

The norm of a transformer parameter set $\boldsymbol{\theta}$ is then defined as $\|\boldsymbol{\theta}\| :=$ $\max\{\text{SelfAttnNorm}, \text{MaskedAttnNorm}, \text{MLPNorm}\}$, where

$$\text{SelfAttnNorm} = \max_{l \in [L_1]} \left\{ \max_{m \in [M]} \left\{ \left\|W_m^{l,Q}\right\|_{\text{op}}, \left\|W_m^{l,K}\right\|_{\text{op}} \right\} + \sum_{m \in [M]} \left\|W_m^{l,V}\right\|_{\text{op}} + \left\|W^{l,O}\right\|_{\text{op}} \right\},$$

$$\text{MaskedAttnNorm} = \max_{l \in [L_2]} \left\{ \max_{m \in [M]} \left\{ \left\|\boldsymbol{Q}_m^{(l)}\right\|_{\text{op}}, \left\|\boldsymbol{K}_m^{(l)}\right\|_{\text{op}} \right\} + \sum_{m \in [M]} \left\|\boldsymbol{V}_m^{(l)}\right\|_{\text{op}} \right\},$$

$$\text{MLPNorm} = \max_{l \in [L_3]} \left\{ \left\|\boldsymbol{W}_1^{(l)}\right\|_{\text{op}} + \left\|\boldsymbol{W}_2^{(l)}\right\|_{\text{op}} \right\},$$

and $\|\cdot\|_{\text{op}}$ is the operator norm.

