# OpenReview forum: "Provable Strategic In-Context Learning of Transformers"
_ICLR.cc/2026/Conference — ICLR 2026 Conference Withdrawn Submission_

### Official Review · Reviewer_FuoU · 2025-10-23

**Soundness:** 2
**Presentation:** 2
**Contribution:** 2
**Rating:** 4
**Confidence:** 3

**Summary:**

This paper proves that Transformers can approximate the online mirror descent (OMD) algorithm in-context to solve matrix and Markov zero-sum games via supervised pre-training. In particular, the method and the corresponding proof remove the reliance on external sampling mechanisms during inference. In the empirical study, the authors show the effect of trajectory length on the model performance.

**Strengths:**

- The paper is overall easy to follow, where the main motivation is easily identifiable.
- The method removes the reliance on external sampling inference during inference time, making the in-context learning more native to the model.
- The paper provides a comprehensive analysis of how a Transformer can approximate the OMD algorithm in-context, as well as how the error would diminish with the trajectory length.

**Weaknesses:**

- The paper only shows that Transformers with sufficient size *can* approximate the OMD algorithm. However, it is insufficient to conclude that OMD is the algorithm the Transformer is implementing in-context. A direct comparison between the trained Transformer and the explicit OMD algorithm, at least at the behavioural level, is missing.
- The empirical study is thin and lacks discussion. I found it difficult to draw any conclusion from Figure 1. It doesn't seem to me that the model performance is increasing with the trajectory length. Therefore, I am unsure if in-context learning is happening at all. The exploitability appears irrelevant to the number of actions, either. I wish the authors could explain the insights they could draw from the experiment results.

Some minor concerns:
- The citation format has an issue. When a citation like xxx et al. is used as part of the sentence, there is no need to repeat it. When it is not a part of the sentence, then a good practice is to wrap it in parentheses.
- Some quantity/notation is used without definition. For example, where is $\epsilon_{approx}$ defined in the main text?
- Some notations are overloaded heavily. For instance, $A$ was used to denote the payoff matrix, the action space, and the linear extraction mapping. It can confuse the readers.
- The paper overlooks some recent related work on how Transformers can approximate RL algorithms in-context. For example, Transformers Can Learn Temporal Difference Methods for In-Context Reinforcement Learning (Wang et al., 2025) studies this connection in depth.

**Questions:**

- In line 123, if $R_G(a,b)$ is a mapping from the joint action space to a scalar, then how can someone sample from it $r_G(a,b) \sim R_G(a,b)$? Where does the stochasticity come from?

---

### Official Review · Reviewer_sBnc · 2025-10-31

**Soundness:** 3
**Presentation:** 2
**Contribution:** 2
**Rating:** 4
**Confidence:** 3

**Summary:**

This paper studies the ability of transformers for in-context game playing in matrix and Markov zero-sum games. The authors show that transformers are able to approximate an online mirror descent (OMD) with $\ell_2$ regularization to efficiently find $\epsilon$-Nash equilibrium in Markov zero-sum games. Moreover, they show that through imitating an expert algorithm in pretraining, the pretrained transformer can achieve comparable performance with OMD. Experimental results on a small Markov game verify their findings on the tradeoff between trajectory length and model performance.

**Strengths:**

This work shows that transformers are able to perform in-context game playing without external sampling procedures as required in previous works. To achieve this, the authors show that transformers are able to approximate a $\ell_2$ regularized OMD, which can be used to find near Nash equilibriums in zero-sum Markov games. I find the theoretical results interesting and provide valuable insights into what kinds of procedures Transformer models can perform.

**Weaknesses:**

The writing of this work is less clear. For example, it is unlcear to me what the expert algorithm is from the main text. Moreover, the bounds in the main theorems (Theorem 4.2, 4.4 ) seem to depend on the approximation error $\epsilon_{real}$, while an explict bound on $\epsilon_{real}$ is not given.

**Questions:**

1. As mentioned above, what are the expert algorithms considered in this work? Moreover, for certain expert algorithms, is it possible to provide concrete bounds on the approximation error $\epsilon_{real}$?

2. What is the relation between Definition 4.1 and those in Shi et al., 2024 and Lin et al., 2024? Is Definition 4.1 a weaker condition?

3. Could the authors discuss the tightness of each term in the main results (Theorems 4.2 and 4.4)? In particular, how do these bounds compare with the convergence rate of online mirror descent? Does OMD with $\ell_2$ regularization achieve the same convergence rate as OMD with KL regularization?

4. Assumption 4.1 is fairly strong, especially since it assumes the division can be approximated for $y \in [0,1]$ instead of $y>c$ for some $c>0$. I wonder what the quantitative result would be when using Transformers to approximate the division operation.

---

### Official Review · Reviewer_JP1w · 2025-11-01

**Soundness:** 3
**Presentation:** 3
**Contribution:** 3
**Rating:** 4
**Confidence:** 3

**Summary:**

The paper investigates whether transformers can provably perform uncoupled no-regret in-context learning (ICL) in competitive games. They consider repeated and Markov zero-sum games under bandit feedback. The previous work has shown that transformers can implement V-learning. The authors highlight the external sampling mechanism that V-learning requires, which is implemented outside the transformer. The main contribution of the paper is that they show transformer can implement the online mirror descent (OMD) algorithm with squared Euclidean norm regularizer. The authors state that this algorithm does not require the external sampling step and has last-iterate convergence guarantees in various game settings. They further mention the previous proof approaches does not extend to the OMD algorithm because a general closed-form solution does not exist. A summary of the theoretical contributions are as follows:
- Theorem 4.1: In zero-sum games, existence of a transformer with sufficient width and depth can implement the OMD algorithm with at most $\epsilon + \eta A$ error for any $\epsilon > 0$, where $\eta$ is the step size for OMD algorithm and $A$ is the number of actions.
- Theorem 4.2: Upper bound exploitability of the pretrained transformer in zero-sum games
- Theorem 4.3: In zero-sum Markov games, existence of a transformer with sufficient width and depth can implement the OMD algorithm with at most $\epsilon + \eta A$ error for any $\epsilon > 0$, where $\eta$ is the step size for OMD algorithm and $A$ is the number of actions.
- Theorem 4.4: Upper bound exploitability of the pretrained transformer in zero-sum Markov games

**Strengths:**

- The authors show transformers' game playing capabilities without external sampling. The results and analysis look technically sound.
- The paper is well-written and easy-to-follow.

**Weaknesses:**

- The error bound realizability result Theorem 4.1 scales with number of actions and the step size. Therefore, as step size grows, the error rate might be high. For comparison, the main related work [1] shows the existence of a transformer that implements V-learning exactly (although it still requires the external sampling mechanism that the authors highlight). The authors also note this in Remark 4.2 but it is not obvious whether this is a proof artifact or an inherent limitation of transformers. It would be helpful if the authors could provide some discussion about why or why not this is a proof artifact.
- The experiments are not convincing. The normalized exploitability of 0.3-0.5 seems too high. The new insight claimed in Remark 4.2 could have been experimentally investigated. There were no discussion related to the experimental results.

Minor issues:
- It looks like there are issues with the reference format throughout the paper. In many cases, paranthetical citations (\citep) should be used instead of the textual ones (\citet); e.g. for lines 34 and 35. When the textual ones should be used (e.g. line 52, double Shi et al.), there is no need to repeat the author name as the \citet puts the name out of the paranthesis.
- In Definition 4.1, one of the $\epsilon_{\mathrm{real}}$'s is redundant.
- Why is the proof of Theorem 4.4 placed before the proof of Theorem 4.3?
- line 366: $\epsilon_{\mathrm{approx}}$ -> $\epsilon_{\mathrm{real}}$
- line 441: lengthsnacross -> lengths across

[1] Shi, C., Yang, K., Yang, J. and Shen, C., 2024. Transformers as Game Players: Provable In-context Game-playing Capabilities of Pre-trained Models. Advances in Neural Information Processing Systems, 37, pp.132001-132049.

**Questions:**

- What do you mean by "We note that this algorithm closely resembles the one used to achieve last-iterate convergence in Markov zero-sum games." (line 399-401)? Is this algorithm only similar to (but not the same as) the OMD algorithm with last-iterate convergence guarantees? Is this about the choice of the regularizer? Does the version in your paper have last-iterate convergence or other guarantees?
- As noted in the weaknesses section, the dependence on $\eta A$ might make the error bound loose. Could you provide the theoretical values and some examples for this error bounds? For example, given a zero-sum Markov game with $A$ actions, what is the highest possible $\eta$ that the analysis of OMD suggests? What is the resulting error bound given by Theorem 4.1 for this game and chosen $\eta$?
- In line 369, it is noted that the $O(T^{-3/8})$ term is due to the last-iterate convergence rate of OMD but there are no such terms in Theorem 4.2? What dies this refer to?
- Do both plots in Figure 1 show the normalized exploitability? The y-axis on the left plot says Normalized Exploitability while the right one says Exploitability.
- Isn't the normalized exploitability values of 0.3-0.5 as Figure 1 shows too high? Could you discuss the experimental results? What is an acceptable value? Assuming that the model you use in your experiments is the one with theoretical guarantees, what are the error bounds given by your theory?

[1] Shi, C., Yang, K., Yang, J. and Shen, C., 2024. Transformers as Game Players: Provable In-context Game-playing Capabilities of Pre-trained Models. Advances in Neural Information Processing Systems, 37, pp.132001-132049.

---

### Official Review · Reviewer_A1Pi · 2025-11-01

**Soundness:** 2
**Presentation:** 2
**Contribution:** 3
**Rating:** 4
**Confidence:** 3

**Summary:**

This paper theoretically investigates whether Transformers can perform in-context game playing in matrix and Markov zero-sum games entirely within their architecture, without external procedures. Authors show the first theoretical result that Transformers can approximate online mirror descent (OMD) dynamics in both repeated and sequential multi-agent games, with the accuracy dependent on model size. Authors demonstrate the benefit of pre-training with longer trajectories under appropriate model architecture choices.

**Strengths:**

1. Authors provide the first theoretical results showing that Transformers can perform in-context learning in competitive games without relying on any external sampling algorithm at inference time. Authors further validate this with empirical results.
2. Authors develop a new approach to analyze Transformers. They show that Transformers can learn in-context by approximating online mirror descent (OMD) dynamics in repeated normal-form games and extend this capability to sequential multi-agent interactions in Markov games. To enable this, we reformulate the constrained OMD updates using the Lagrangian method, allowing Transformers to approximate the resulting unconstrained optimization dynamics. Instead of relying on smooth or approximation through gradient descent, developed approach bridges constrained dynamics and transformer architectures through a careful analysis of the smooth surrogate objective and its approximation error.
3. Demonstrated results show that while Transformers can approximate OMD up to an arbitrary ϵ-accuracy, doing so imposes architectural constraints.
4. Authors extend the results to Markov zero-sum games. While the previous error bound scales linearly with the trajectory length T, which suggests a less efficient training with longer trajectories, our results show the opposite that the duality gap in in-context game playing decreases as T increases, provided the Transformer architecture is sufficiently large.

**Weaknesses:**

1) The article does not contain a detailed experimental verification of the theoretical results obtained; only one experiment was carried out for a particular example.

2) There is no quantitative or qualitative comparison of the proposed approach with existing similar methods.

**Questions:**

1) Can the developed approach be used in more complex gaming environments, including simulators?

---

### Note · Authors · 2025-11-18

**Comment:**

We thank the reviewer for the constructive comments, and we will work on improving the work for a future venue.

**Withdrawal Confirmation:**

I have read and agree with the venue's withdrawal policy on behalf of myself and my co-authors.